# Malaria protection due to sickle haemoglobin depends on parasite genotype

Gavin Band[1,2,3 ✉], Ellen M. Leffler[2,4], Muminatou Jallow[5,6], Fatoumatta Sisay-Joof[5], Carolyne M. Ndila[7], Alexander W. Macharia[7], Christina Hubbart[1], Anna E. Jeffreys[1], Kate Rowlands[1], Thuy Nguyen[2], Sónia Gonçalves[2], Cristina V. Ariani[2], Jim Stalker[2], Richard D. Pearson[2,3], Roberto Amato[2], Eleanor Drury[2], Giorgio Sirugo[5,8], Umberto d'Alessandro[5], Kalifa A. Bojang[5], Kevin Marsh[7,9], Norbert Peshu[7], Joseph W. Saelens[10], Mahamadou Diakité[11], Steve M. Taylor[10,12], David J. Conway[5,13], Thomas N. Williams[7,14], Kirk A. Rockett[1,2 ✉] & Dominic P. Kwiatkowski[1,2,3 ✉]

Host genetic factors can confer resistance against malaria[1], raising the question of whether this has led to evolutionary adaptation of parasite populations. Here we searched for association between candidate host and parasite genetic variants in 3,346 Gambian and Kenyan children with severe malaria caused by *Plasmodium falciparum*. We identified a strong association between sickle haemoglobin (HbS) in the host and three regions of the parasite genome, which is not explained by population structure or other covariates, and which is replicated in additional samples. The HbS-associated alleles include nonsynonymous variants in the gene for the acyl-CoA synthetase family member[2–4] *PfACS8* on chromosome 2, in a second region of chromosome 2, and in a region containing structural variation on chromosome 11. The alleles are in strong linkage disequilibrium and have frequencies that covary with the frequency of HbS across populations, in particular being much more common in Africa than other parts of the world. The estimated protective effect of HbS against severe malaria, as determined by comparison of cases with population controls, varies greatly according to the parasite genotype at these three loci. These findings open up a new avenue of enquiry into the biological and epidemiological significance of the HbS-associated polymorphisms in the parasite genome and the evolutionary forces that have led to their high frequency and strong linkage disequilibrium in African *P. falciparum* populations.

Malaria can be viewed as an evolutionary arms race between the host and parasite populations. Human populations in Africa have acquired a high frequency of HbS and other erythrocyte polymorphisms that provide protection against the severe symptoms of *P. falciparum* infection[1,5], while *P. falciparum* populations have evolved a complex repertoire of genetic variation to evade the human immune system and to resist antimalarial drugs[6,7]. This raises the basic question: are there genetic forms of *P. falciparum* that can overcome the human variants that confer resistance to this parasite?

To address this question, we analysed both host and parasite genome variation in samples from 5,096 children from Gambia and Kenya with severe malaria caused by *P. falciparum* (Extended Data Fig. 1, Supplementary Fig. 1, Methods). The samples were collected over the period 1995–2009 as part of a genome-wide association study (GWAS) of human resistance to severe malaria[5,8,9]. In brief, we sequenced the *P. falciparum* genome using the Illumina X Ten platform using two approaches based on sequencing whole DNA and selective whole-genome amplification[10]. We used an established pipeline[11] to identify and call genotypes at more than two million single nucleotide polymorphisms (SNPs) and short insertion and deletion variants across the *P. falciparum* genome in these samples (Methods), although the majority of these occurred at low frequency. Our analysis is based on the 4,171 samples that had high quality data for both parasite and human genotypes, of which a subset of 3,346 had human genome-wide genotyping available and were used for discovery analysis. We focused on a set of 51,225 biallelic variants in the *P. falciparum* genome that passed all quality control filters and were observed in at least 25 infections in this subset (Methods). Our analyses exclude mixed-genotype calls

[1]Wellcome Centre for Human Genetics, University of Oxford, Oxford, UK. [2]Wellcome Sanger Institute, Hinxton, Cambridge, UK. [3]Big Data Institute, Li Ka Shing Centre for Health and Information Discovery, University of Oxford, Oxford, UK. [4]Department of Human Genetics, University of Utah School of Medicine, Salt Lake City, UT, USA. [5]Medical Research Council Unit The Gambia at the London School of Hygiene and Tropical Medicine, Fajara, The Gambia. [6]Edward Francis Small Teaching Hospital (formerly Royal Victoria Teaching Hospital), Banjul, The Gambia. [7]KEMRI-Wellcome Trust Research Programme, Kilifi, Kenya. [8]Division of Translational Medicine and Human Genetics, University of Pennsylvania School of Medicine, Philadelphia, PA, USA. [9]Nuffield Department of Medicine, University of Oxford, Oxford, UK. [10]Division of Infectious Diseases, Duke University School of Medicine, Durham, NC, USA. [11]Malaria Research and Training Center, University of Sciences, Techniques, and Technologies of Bamako, Bamako, Mali. [12]Duke Global Health Institute, Duke University, Durham, NC, USA. [13]Faculty of Infectious and Tropical Diseases, London School of Hygiene and Tropical Medicine, London, UK. [14]Institute for Global Health Innovation, Department of Surgery and Cancer, Imperial College London, London, UK. ✉e-mail: gavin.band@well.ox.ac.uk; kirk.rockett@well.ox.ac.uk; dominic@sanger.ac.uk

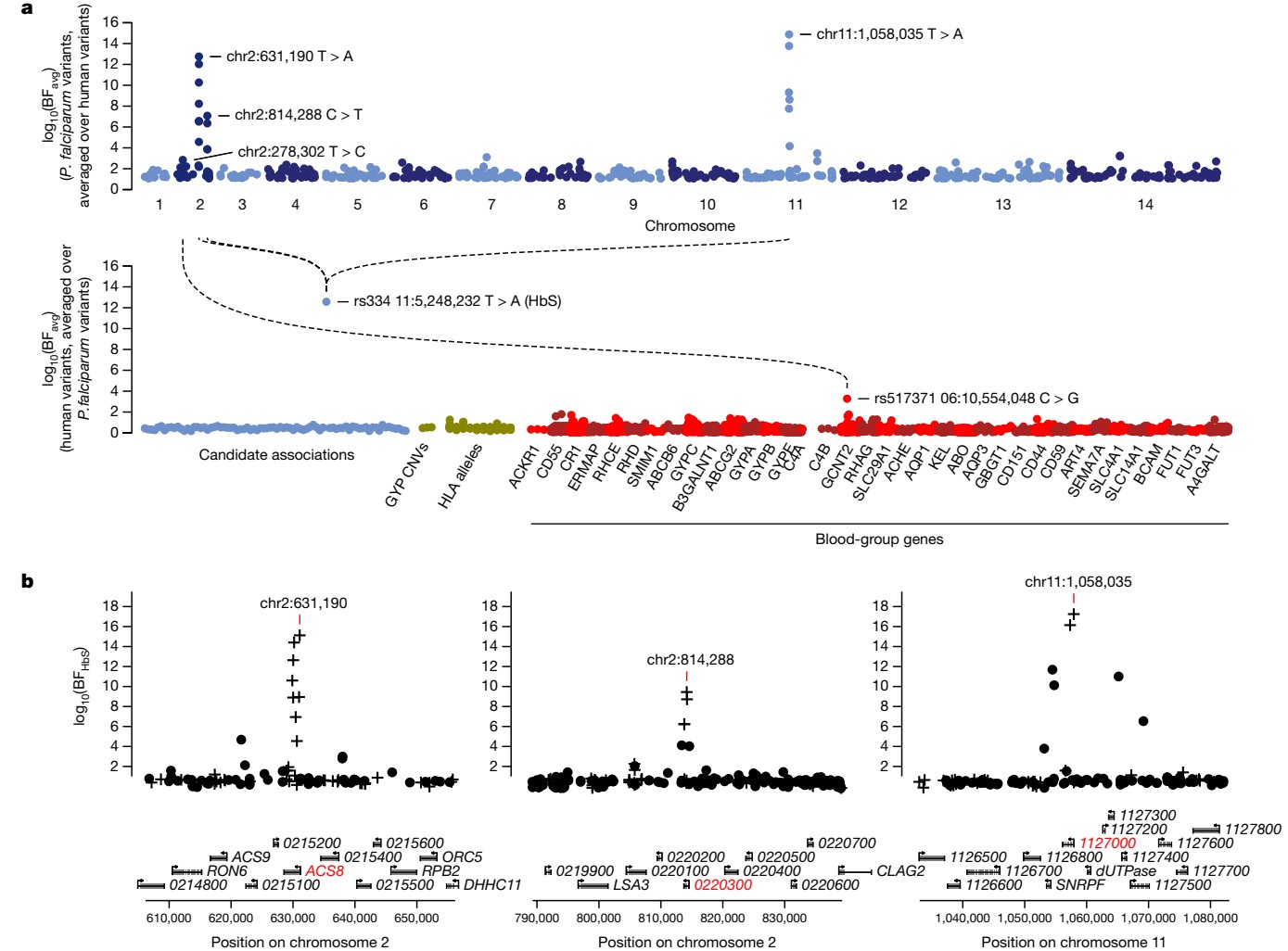

**Fig. 1 | Three regions of the *P. falciparum* genome are associated with HbS.** **a**, Points show the evidence for association between each *P. falciparum* variant and human genotypes (top row) or between each included human variant and *P. falciparum* genotypes (bottom row). Association evidence is summarized by averaging the evidence for pairwise association (Bayes factor (BF) for test in $n$ = 3,346 samples) between each variant (points) and all variants in the other organism against which it was tested ($\log_{10}(BF_{avg})$). *P. falciparum* variants are shown grouped by chromosome, and human variants are grouped by inclusion category as described in text and Methods. Dashed lines and variant annotations reflect pairwise tests with BF > 10⁶; only the top signal in each association region pair is annotated (Methods). **b**, Detail of the association with HbS in the *Pfsa1*, *Pfsa2* and *Pfsa3* regions of the *P. falciparum* genome. Points show evidence for association with HbS ($\log_{10}(BF_{HbS})$) for each regional variant. Variants that alter protein coding sequence are denoted by plus, and other variants are denoted by circles. Results are computed by logistic regression including an indicator of country as a covariate and assuming an additive model of association, with HbS genotypes based on imputation from genome-wide genotypes as previously described[8]. Mixed and missing *P. falciparum* genotype calls were excluded from the computation. Below, regional genes are annotated, with gene symbols given where the gene has an ascribed name in the PlasmoDB annotation (after removing 'PF3D7_' from the name where relevant); the three genes containing the most-associated variants are shown in red. A corresponding plot using directly typed HbS genotypes is presented in Extended Data Fig. 2.

that arise in malaria when a host is infected with multiple parasite lineages. Full details of our sequencing and data processing can be found in Supplementary Methods.

We used a logistic-regression approach to test for pairwise association between these *P. falciparum* variants and four categories of human variants that are plausibly associated with malaria resistance: (1) known autosomal protective mutations, including HbS (in *HBB*), the common mutation that determines the O blood group (in *ABO*), regulatory variation associated with protection at *ATP2B4*[5,8,12] and the structural variant DUP4, which encodes the Dantu blood-group phenotype[13]; (2) variants that showed suggestive but not conclusive evidence for association with severe malaria in our previous GWAS[8]; (3) human leukocyte antigen (HLA) alleles and additional glycophorin structural variants that we previously imputed in these samples[8,13]; and (4) variants near genes that encode human blood-group antigens, which we

tested against the subset of *P. falciparum* variants lying near genes that encode proteins important for the merozoite stage[14,15], as these might conceivably interact during host cell invasion by the parasite. Although several factors could confound this analysis in principle—notably, if there was incidental association between human and parasite population structure—the distribution of test statistics suggested that our test was not affected by systematic confounding after including only an indicator of country as a covariate (Supplementary Fig. 2), and we used this approach for our main analysis. The full set of results is summarized in Fig. 1a, Supplementary Table 1.

## Three *P. falciparum* loci are associated with HbS

The most prominent finding to arise from this joint analysis of host and parasite variation was a strong association between the sickle haemoglobin allele HbS and three separate regions in the *P. falciparum* genome

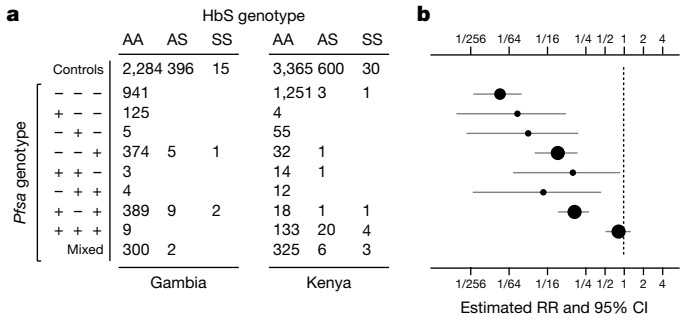

**a**

| | HbS genotype | | | | | |
|---|---|---|---|---|---|---|
| | AA | AS | SS | AA | AS | SS |
| Controls | 2,284 | 396 | 15 | 3,365 | 600 | 30 |
| − − − | 941 | | | 1,251 | 3 | 1 |
| + − − | 125 | | | 4 | | |
| − + − | 5 | | | 55 | | |
| − − + | 374 | 5 | 1 | 32 | 1 | |
| + + − | 3 | | | 14 | 1 | |
| − + + | 4 | | | 12 | | |
| + − + | 389 | 9 | 2 | 18 | 1 | 1 |
| + + + | 9 | | | 133 | 20 | 4 |
| Mixed | 300 | 2 | | 325 | 6 | 3 |
| | Gambia | | | Kenya | | |

(*Pfsa* genotype labels the rows; the leftmost axis is labelled *Pfsa* genotype)

**b** (scale: 1/256 1/64 1/16 1/4 1/2 1 2 4) — Estimated RR and 95% CI

**Fig. 2 | The estimated relative risk for HbS varies by *Pfsa* genotype.**
**a**, Numbers of cases of severe malaria from the Gambia and Kenya with indicated HbS genotype (columns) and carrying the indicated alleles at the *Pfsa1*, *Pfsa2* and *Pfsa3* loci (rows; using $n = 4,054$ samples with directly typed HbS genotype and non-missing genotype at the three *P. falciparum* loci). *Pfsa* alleles positively associated with HbS are denoted + and those negatively associated with HbS are denoted − for the respective loci. Samples with mixed *P. falciparum* genotype calls for at least one of the loci are shown in the bottom row and further detailed in Extended Data Fig. 4. The first row indicates counts of HbS genotypes in population control samples from the same populations[8]. **b**, The estimated relative risk of HbS for severe malaria with *Pfsa* genotypes (rows) as indicated in **a**. Relative risks were estimated using a multinomial logistic regression model with controls as the baseline outcome and assuming complete dominance (that is, that HbAS and HbSS genotypes have the same association with parasite genotype) as described in Supplementary Methods; an indicator of country was included as a covariate. Circles reflect posterior mean estimates and horizontal lines reflect the corresponding 95% credible intervals (CI). Estimates based on less than 5 individuals with HbAS or HbSS genotypes are represented by smaller circles. To reduce overfitting we used Stan[46] to fit the model assuming a mild regularising Gaussian prior with mean zero and standard deviation of 2 on the log-odds scale (that is, with 95% of mass between 1/50 and 50 on the relative risk scale) for each parameter, and between-parameter correlations set to 0.5.

(Fig. 1b). Additional associations with marginal levels of evidence were observed at a number of other loci, including a potential association between *GCNT2* in the host and *PfMSP4* in the parasite and associations involving HLA alleles (detailed in Supplementary Methods, Supplementary Table 1), but here we focus on the association with HbS.

The statistical evidence for association at the HbS-associated loci can be described as follows, focussing on the variant with the strongest association in each region and assuming an additive model of effect of the host allele on parasite genotype on the log-odds scale (Supplementary Table 1). The chr2: 631,190 T>A variant, which lies in *PfACS8*, was associated with HbS with a Bayes factor ($BF_{HbS}$) of $1.1 \times 10^{15}$ (computed under a log $F(2,2)$ prior; Methods) and $P$ value of $4.8 \times 10^{-13}$ (computed using a Wald test; Supplementary Methods). At a second region on chromosome 2, the chr2: 814,288 C>T variant, which lies in *Pf3D7_0220300*, was associated with $BF_{HbS} = 2.4 \times 10^9$ and $P = 1.6 \times 10^{-10}$. At the chromosome 11 locus, the chr11: 1,058,035 T>A variant, which lies in *Pf3D7_1127000*, was associated with $BF_{HbS} = 1.5 \times 10^{17}$ and $P = 7.3 \times 10^{-12}$. For brevity, we refer to these HbS-associated loci as *Pfsa1*, *Pfsa2* and *Pfsa3*, respectively—for *P. falciparum* sickle-associated—and we use + and − signs to refer to alleles that are positively and negatively correlated with HbS, respectively. For example, *Pfsa1⁺* denotes the allele that is positively correlated with HbS at the *Pfsa1* locus. All three of the lead variants are nonsynonymous mutations of their respective genes, as are additional associated variants in these regions (Fig. 1, Supplementary Table 1).

The above results are based on HbS genotypes imputed from surrounding haplotype variation[8], but we focus below on the larger set of 4,071 cases in which we have previously directly assayed HbS genotypes[5] (Extended Data Fig. 1). This includes the majority of samples used in our discovery analysis. The *Pfsa1* and *Pfsa3* associations were

clearly supported in both populations in this dataset, whereas *Pfsa2⁺* appears rare in Gambia (Supplementary Tables 2, 3). We also observed convincing replication of the associations in the additional 825 samples that were not part of our discovery phase, with nominal replication of *Pfsa3* in the Gambia (one-tailed $P = 0.026$, $N = 163$) and replication of all three loci in the larger sample from Kenya ($P < 0.001$, $N > 540$) (Supplementary Table 2). Across the full dataset there is thus very strong evidence of association with HbS at all three loci ($BF_{HbS} = 2.0 \times 10^{21}$ for *Pfsa1*, $3.7 \times 10^{12}$ for *Pfsa2*, and $1.4 \times 10^{24}$ for *Pfsa3*; Extended Data Fig. 2) with corresponding large effect size estimates (estimated odds ratio (OR) = 12.8 for *Pfsa1⁺*, 7.5 for *Pfsa2⁺* and 21.7 for *Pfsa3⁺*). As described above, these estimates assume an additive relationship between HbS and the *P. falciparum* genotype at each locus, but we also noted that there is greatest evidence for a dominance effect (Supplementary Tables 2, 3).

We further examined the effect of adjusting for covariates in our data, including human and parasite principal components reflecting population structure, year of sampling, clinical type of severe malaria and technical features related to sequencing (Extended Data Fig. 3). Inclusion of these covariates did not substantially affect results with one exception: we found that parasite principal components computed across the whole *P. falciparum* genome included components that correlated with the *Pfsa* loci, and including these principal components reduced the association signal, particularly in Kenya. Altering the principal components by removing the *Pfsa* regions restored the association, indicating that this is not caused by a general population structure effect that is reflected in genotypes across the parasite genome, and we further discuss the reasons for this finding below. Finally, we analysed available data from a set of 32 uncomplicated infections of Malian children ascertained based on HbS genotypes[16] (Methods); this provided further replication of the associations with *Pfsa1* and *Pfsa3* (Supplementary Table 2). Together, these data indicate that there are genuine differences in the distribution of parasite genotypes between infections of HbS and non-HbS genotype individuals.

## HbS protection varies with parasite type

The level of protection afforded by HbS against severe malaria can be estimated by comparing its frequency between cases and population controls. As shown in Fig. 2, the vast majority of children with HbS genotype in our data were infected with parasites that carry *Pfsa⁺* alleles. Corresponding to this, our data show little evidence of a protective effect of HbS against severe malaria with parasites of *Pfsa1⁺*, *Pfsa2⁺* and *Pfsa3⁺* genotype (estimated relative risk (RR) = 0.83, 95% confidence interval = 0.53–1.30). By contrast, HbS is strongly associated with reduced risk of disease caused by parasites of *Pfsa1⁻*, *Pfsa2⁻* and *Pfsa3⁻* genotype (RR = 0.01, 95% confidence interval = 0.007–0.03). These estimates should be interpreted with caution because they are based on just 49 cases of severe malaria that had an HbS genotype, because many of these samples were included in the initial discovery dataset, and because there is some variation evident between populations. However, it can be concluded that the protective effect of HbS is dependent on parasite genotype at the *Pfsa* loci.

## Population genetics of the *Pfsa* loci

The *Pfsa1⁺*, *Pfsa2⁺* and *Pfsa3⁺* alleles had similar frequencies in Kenya (approximately 10–20%) whereas in Gambia *Pfsa2⁺* had a much lower allele frequency than *Pfsa1⁺* or *Pfsa3⁺* (below 3% in all years studied, versus 25–60% for the *Pfsa1⁺* or *Pfsa3⁺* alleles; Fig. 3a). To explore the population genetic features of these loci in more detail, we analysed the MalariaGEN Pf6 open resource, which provides *P. falciparum* genome variation data for 7,000 worldwide samples[11] (Fig. 3b). This showed considerable variation in the frequency of these alleles across Africa, the maximum observed value being 61% for *Pfsa3⁺* in the Democratic Republic of Congo, and indicated that these alleles are rare outside Africa. Moreover, we found that within Africa, population frequencies

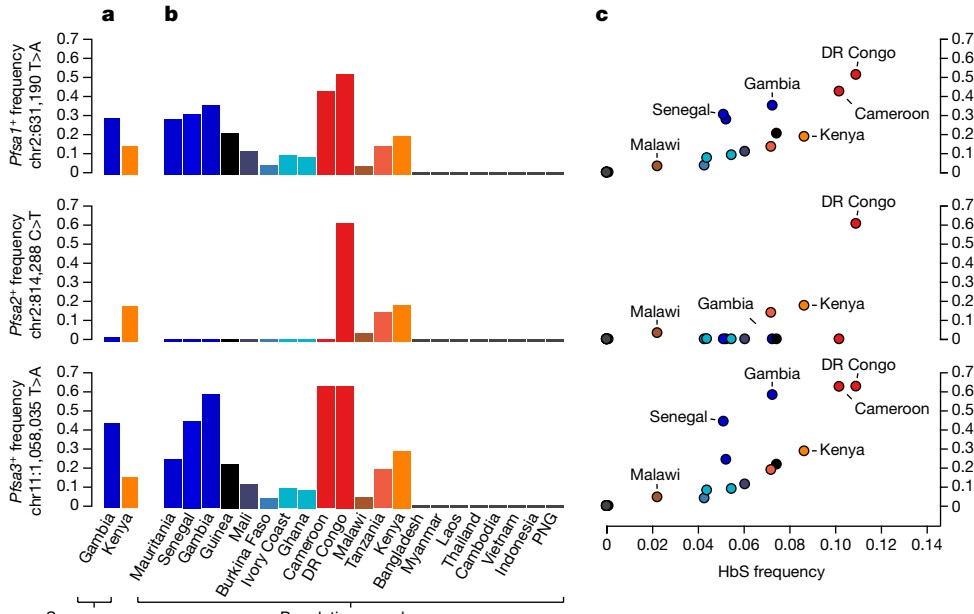

**Fig. 3 | The relationship between *Pfsa* and HbS allele frequencies across populations. a**, Bars show the estimated frequency of each *Pfsa*+ allele in severe cases of malaria from each country. Details of allele frequencies and sample counts across years are presented in Extended Data Fig. 5. **b**, Estimated frequency of each *Pfsa*+ allele in worldwide populations from the MalariaGEN Pf6 resource[11], which contains samples collected during the period 2008–2015. Only countries with at least 50 samples are shown (this excludes Columbia, Peru, Benin, Nigeria, Ethiopia, Madagascar and Uganda). **c**, Estimated population-level *Pfsa*+ allele frequency (as in **a**, **b**) against HbS allele frequency in populations from MalariaGEN Pf6 (coloured as in **b**; selected populations are also labelled). *Pfsa*+ allele frequencies were computed from the relevant genotypes, after excluding mixed or missing genotype calls. HbS allele frequencies were computed from frequency estimates previously published by the Malaria Atlas Project[17] for each country, by averaging over the locations of MalariaGEN Pf6 sampling sites weighted by the sample size. DR, Democratic Republic; PNG, Papua New Guinea.

of the *Pfsa*+ alleles are strongly correlated with the frequency of HbS (Fig. 3c; estimated using data from the Malaria Atlas Project[17]).

This analysis also revealed a further feature of the *Pfsa*+ alleles: although *Pfsa1* and *Pfsa2* are separated by 180 kb, and the *Pfsa3* locus is on a different chromosome, they are in strong linkage disequilibrium (LD). This can be seen from the co-occurrence of these alleles in severe cases (Fig. 2), and from the fact that they covary over time in our sample (Extended Data Fig. 5) and geographically across populations (Fig. 3b). We computed LD metrics between the *Pfsa*+ alleles in each population (Supplementary Table 4) after excluding HbS-carrying individuals to avoid confounding with the association outlined above. *Pfsa1*+ and *Pfsa2*+ were strongly correlated in Kenyan severe cases ($r = 0.75$) and *Pfsa1*+ and *Pfsa3*+ were strongly correlated in both populations ($r = 0.80$ in Kenya; and $r = 0.43$ in severe cases from the Gambia). This high LD was not explained by population structure or other covariates in our data (Methods), and was also observed in multiple populations in MalariaGEN Pf6 (for example, $r = 0.20$ between *Pfsa1*+ and *Pfsa3*+ in the Gambia; $r = 0.71$ in Kenya; and $r > 0.5$ in all other African populations surveyed; Supplementary Table 4), showing that the LD is not purely an artefact of our sample of severe malaria cases.

This observation of strong correlation between alleles at distant loci is unexpected, because the *P. falciparum* genome undergoes recombination in the mosquito vector and typically shows very low levels of LD in malaria-endemic regions[11,18,19]. To confirm that this is unusual, we compared LD between the *Pfsa* loci with the distribution computed from all common biallelic variants on different chromosomes (Fig. 4). In Kenyan samples, the *Pfsa* loci have the highest between-chromosome LD of any pair of variants in the genome. In Gambia, between-chromosome LD at these SNPs is also extreme, but another pair of extensive regions on chromosomes 6 and 7 also show strong LD. These regions contain the chloroquine resistance-linked genes *PfCRT* and *PfAAT1*[20,21] and contain long stretches of DNA sharing identical by descent, consistent with positive selection of antimalarial-resistant

haplotypes[22]. Moreover, we noted that these signals are among a larger set of HbS-associated and drug-resistance loci that appear to have increased between-chromosome LD in these data (Supplementary Table 4).

Combining these new findings with other population genetic evidence from multiple locations across Africa, including observations of frequency differentiation within and across *P. falciparum* populations[11,23,24] and other metrics at these loci indicative of selection[22,25,26], it appears likely that the allele frequencies and strong LD between *Pfsa1*, *Pfsa2* and *Pfsa3* are maintained by some form of natural selection. However, the mechanism for this is unclear. Given our findings, an obvious hypothesis is that the *Pfsa1*+, *Pfsa2*+ and *Pfsa3*+ alleles are positively selected in hosts with HbS, but since the frequency of HbS carriers[5,17] is typically <20% it is not clear whether this alone is a sufficient explanation to account for the high population frequencies or the strong LD observed in non-HbS carriers. Equally, since the *Pfsa*+ alleles have not reached fixation (Fig. 3) and do not appear to be rapidly increasing in frequency (Extended Data Fig. 5), an opposing force may also be operating to maintain their frequency. However, the above data do not suggest strong fitness costs for *Pfsa*+-carrying parasites in HbAA individuals (Fig. 2), and the *Pfsa2*+ allele also appears to be present only in east Africa, further complicating these observations. It thus remains entirely possible that additional selective factors are involved, such as epistatic interactions between these loci, or further effects on fitness in the host or vector in addition to those observed here in relation to HbS.

### The genomic context of the *Pfsa* variants
The biological function of these parasite loci is an area of considerable interest for future investigation. At the *Pfsa1* locus, the signal of association includes non-synonymous changes in the *PfACS8* gene, which encodes an acyl-CoA synthetase[3] that belongs to a gene family that has expanded in the Laverania relative to other *Plasmodium* species[4]

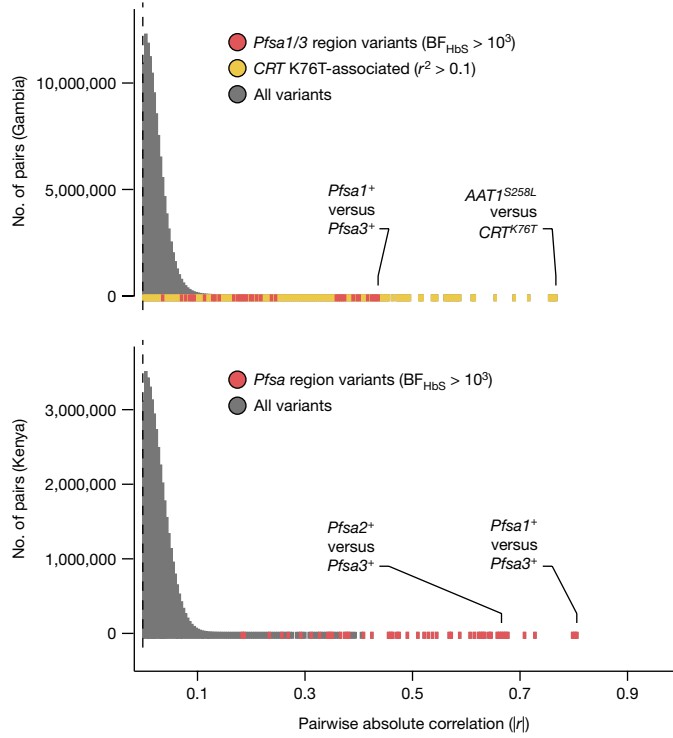

**Fig. 4 | HbS-associated variants show extreme between-chromosome correlation in severe *P. falciparum* infections.** Empirical distribution of absolute genotype correlation ($|r|$) between pairs of variants on different *P. falciparum* chromosomes in the Gambia (top) and Kenya (bottom). To avoid capturing direct effects of the HbS association, correlation values are computed after excluding HbS-carrying individuals. All pairs of biallelic variants with estimated minor allele frequency at least 5% and at least 75% of samples having non-missing and non-mixed genotype call are shown (totalling 16,487 variants in the Gambia and 13,766 variants in Kenya). Colours indicate the subset of comparisons between HbS-associated variants in *Pfsa* regions relevant for the population (red) and between variants in LD with the CRT K76T mutation. Labelled points denote the variant pairs showing the highest and second-highest pairwise correlation in each population after grouping correlated variants into regions; for this purpose regions were defined to include all nearby pairs of correlated variants with minor allele frequency ≥5% and $r^2 > 0.05$, such that no other such pair of variants within 10 kb of the given region boundaries is present (Methods). A longer list of regions showing increased between-chromosome LD is presented in Supplementary Table 5.

and lies close to a paralogue *PfACS9* on chromosome 2. *PfACS8* has been predicted to localize to the apicoplast[27], but it also contains a *Plasmodium* export element (PEXEL)-like motif[28–30], which may instead indicate export to the host cytosol where other acyl-CoA synthetase family members have been observed[31]. The functions of the proteins encoded by *PF3D7_0220300* (an exported protein, at the *Pfsa2* locus) and *PF3D7_1127000* (a putative tyrosine phosphatase, at *Pfsa3*) are not known; however, the protein encoded by *PF3D7_0220300* has been observed to localise to the host membrane and to colocalise with host stomatin[32], whereas the protein encoded by *PF3D7_1127000* has been observed in the food vacuole[33]. All three genes appear to be expressed at multiple parasite lifecycle stages (Supplementary Text) in 3D7 parasites and are not essential for in vitro growth[34].

We noted two further features that may point to the functional role of the *Pfsa+* alleles themselves. The associated variants at *Pfsa2* and *Pfsa3* each include SNPs immediately downstream of a PEXEL motif (detailed in Supplementary Information), which mediates export through a pathway that involves protein cleavage at the motif[35]. This process leaves the downstream amino acids at the N terminus of the mature protein, and it is therefore possible that these variants influence successful

export[36,37]. However, another possibility is that the *Pfsa+* alleles affect levels of transcription of the relevant genes. In this context, we noted a recent study[16] that found that *PF3D7_1127000* is among the most differentially over-expressed genes in trophozoite-stage infections of children with HbAS compared with those with HbAA (more than 32-fold increase in transcripts per million (TPM) at the trophozoite stage; $n = 12$; unadjusted $P = 5.6 \times 10^{-22}$). We reanalysed these data in light of genotypes at the *Pfsa* loci (Supplementary Table 6), and found that the *Pfsa3+* mutations plausibly explain this increased expression. In particular, read ratios at the second-most-associated *Pfsa3* SNP (chr11:1,057,437 T > C) (Supplementary Table 1) appear especially strongly correlated with increased expression at trophozoite stage (Extended Data Fig. 6). Further support for this observation comes from an in vitro time-course experiment conducted in the same study[16], in which the increased expression was also observed in HbAA erythrocytes infected with a *Pfsa+*-carrying isolate (Extended Data Figs. 7, 8, Methods). The mechanism of upregulation is not known, but a further relevant observation is that the *Pfsa3+* alleles appear to be linked to a neighbouring copy number variant that includes duplication of the 5′ end of the small nuclear ribonucleoprotein gene *SNRPF*, upstream of *PF3D7_1127000* (based on analysis of available genome assemblies of *P. falciparum* isolates[38]; Extended Data Fig. 9, Supplementary Fig. 3). We caution that these findings are tentative, and the manner in which *Pfsa* alleles affect genome function is a subject for future research. Understanding this functional role could provide important clues into how HbS protects against malaria and help to distinguish between the various proposed mechanisms, which include enhanced macrophage clearance of infected erythrocytes[39], inhibition of intraerythrocytic growth dependent on oxygen levels[40], altered cytoadherence of infected erythrocytes[41] due to cytoskeleton remodelling[42], and immune-mediated mechanisms[43].

## Discussion

A fundamental question in the biology of host–pathogen interactions is whether the genetic makeup of infections is determined by the genotype of the host. While there is some previous evidence of this in malaria—for example, allelic variants of the *PfCSP* gene have been associated with HLA type[44] and HbS has itself previously been associated with MSP-1 alleles[45] (described further in Supplementary Information)—our findings provide clear evidence of an interaction between genetic variants in the parasite and the host. Our central discovery is that among African children with severe malaria there is a strong association between HbS in the host and three loci in different regions of the parasite genome. Based on estimation of relative risk, HbS has no apparent protective effect against severe malaria in the presence of the *Pfsa1+*, *Pfsa2+* and *Pfsa3+* alleles. These alleles, which are much more common in Africa than elsewhere, are positively correlated with HbS allele frequencies across populations. However, they are also found in substantial numbers of individuals without HbS, reaching up to 60% allele frequency in some populations. The *Pfsa1*, *Pfsa2* and *Pfsa3* loci also show remarkably high levels of long-range between-locus LD relative to other loci in the *P. falciparum* genome, which is equally difficult to explain without postulating ongoing evolutionary selection. Although it seems clear that HbS has a key role in this selective process, there is a need for further population surveys (that include asymptomatic and uncomplicated cases of malaria) to gain a more detailed understanding of the genetic interactions between HbS and these parasite loci, and how they affect the overall protective effect of HbS against severe malaria.

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

## Methods

### Ethics and consent

Sample collection and design of our case-control study[8] was approved by Oxford University Tropical Research Ethics committee (OXTREC), Oxford, United Kingdom (OXTREC 020-006). Informed consent was obtained from parents or guardians of patients with malaria, and from mothers for population controls. Local approving bodies were the MRC/Gambia Government Ethics Committee (SCC 1029v2 and SCC670/630) and the KEMRI Research Ethics Committee (SCC1192).

### Building a combined dataset of human and *P. falciparum* genotypes for severe cases

We used Illumina sequencing to generate two datasets jointly reflecting human and *P. falciparum* genetic variation, using a sample of severe malaria cases from the Gambia and Kenya for which human genotypes have previously been reported[5,8]. A full description of our sequencing and data processing is given in Supplementary Methods and summarized in Extended Data Fig. 1. In brief, following a process of sequence data quality control and merging across platforms, we generated (1) a dataset of microarray and imputed human genotypes, and genome-wide *P. falciparum* genotypes, in 3,346 individuals previously identified as without close relationships[8]; and (2) a dataset of HbS genotypes directly typed on the Sequenom iPLEX Mass-Array platform (Agena Biosciences)[5], and genome-wide *P. falciparum* genotypes, in 4,071 individuals without close relationships[8]. Parasite DNA was sequenced from whole DNA in samples with high parasitaemia, and using selective whole-genome amplification (SWGA) to amplify *P. falciparum* DNA in all samples. *P. falciparum* genotypes were called using an established pipeline[11] based on GATK, which calls single nucleotide polymorphisms and short insertion–deletion variants relative to the Pf3D7 reference sequence. This pipeline deals with mixed infections by calling parasite variants as if the samples were diploid; in practice this means that variants with substantial numbers of reads covering reference and alternate alleles are called as heterozygous genotypes.

For the analyses presented in main text, we used the 3,346 samples with imputed human genotypes for our initial discovery analysis, and the 4,071 individuals with directly typed HbS genotypes for all other analysis. The individuals in these two datasets substantially overlap (Extended Data Fig. 1), but a subset of 825 individuals have directly typed for HbS but were not in the discovery data and we used these for replication.

### Inference of genetic interaction from severe malaria cases

To describe our approach, we first consider a simplified model of infection in which parasites have a single definite (measurable) genotype, acquired at time of biting, that is relevant to disease outcome—that is, we neglect any effects of within-host mutation, co- and super-infection at the relevant genetic variants. We consider a population of individuals who are susceptible to being been bitten by an infected mosquito. A subset of infections go on to cause severe disease. Among individuals who are bitten and infected with a particular parasite type $I = x$, the association of a human allele $E = e$ with disease outcome can be measured by the relative risk,

$$RR = \frac{P(\text{disease}|E = e, I = x)}{P(\text{disease}|E = 0, I = x)} \qquad (1)$$

where we have used $E = 0$ to denote a chosen baseline human genotype against which risks are measured. If the strength of association further varies between parasite types (say between $I = x$ and a chosen infection type $I = 0$) then these relative risks will vary, and thus the ratio of relative risks (RRR) will differ from 1. If the host genotype $e$ confers protection against severe malaria, the ratio of relative risks will therefore capture variation in the level of protection compared between different parasite types.

Although phrased above in terms of a relative risk for human genotypes, the RRR can be equivalently expressed as a ratio of relative risks for a given parasite genotype compared between two human genotypes (Supplementary Methods). It is thus conceptually symmetric with respect to human and parasite alleles, and would equally well capture variation in the level of pathogenicity conferred by a particular parasite type compared between different human genotypes.

The OR for specific human and parasite alleles computed in severe malaria cases is formally similar to the ratio of relative risks but with the roles of the genotypes and disease status interchanged. We show in Supplementary Methods that in fact

$$OR = RRR \times OR^{\text{biting}} \qquad (2)$$

where $OR^{\text{biting}}$ is a term that reflects possible association of human and parasite genotypes at the time of mosquito biting. Thus, under this model and in the absence of confounding factors, $OR \neq 1$ implies either that host and parasite genotypes are not independent at time of biting, or that there is an interaction between host and parasite genotypes in determining disease risk. The former possibility may be considered less plausible because it would seem to imply that relevant host and parasite genotypes can be detected by mosquitos prior to or during biting, but we stress that this cannot be tested formally without data on mosquito-borne parasites. A further discussion of these assumptions can be found in Supplementary Methods.

### Testing for genome-to-genome correlation

We developed a C++ program (HPTEST) to efficiently estimate the odds ratio (equation (2)) across multiple human and parasite variants, similar in principle to approaches that have been developed for human-viral and human-bacterial GWAS[47–49]. HPTEST implements a logistic regression model in which genotypes from one file are included as the outcome variable and genotypes from a second file on the same samples are included as predictors. Measured covariates may also be included, and the model accounts for uncertainty in imputed predictor genotypes using the approach from SNPTEST[50]. The model is fit using a modified Newton-Raphson with line search method. For our main analysis we applied HPTEST with the parasite genotype as outcome and the host genotype as predictor, assuming an additive effect of the host genotype on the log-odds scale, and treating parasite genotype as a binary outcome (after excluding mixed and missing genotype calls.).

To mitigate effects of finite sample bias, we implemented regression regularised by a weakly informative log $F(2,2)$ prior distribution[51] on the effect of the host allele (similar to a Gaussian distribution with standard deviation 1.87; Supplementary Methods). Covariate effects were assigned a log $F(0.08,0.08)$ prior, which has similar 95% coverage interval to a gaussian with zero mean and standard deviation of 40. We summarised the strength of evidence using a Bayes factor against the null model that the effect of the host allele is zero. A *P*-value can also be computed under an asymptotic approximation by comparing the maximum posterior estimate of effect size to its expected distribution under the null model (Supplementary Methods). For our main results we included only one covariate, an indicator of the country from which the case was ascertained (Gambia or Kenya); additional exploration of covariates is described below.

### Choice of genetic variants for testing

For our initial discovery analysis we concentrated on a set of 51,552 *P. falciparum* variants that were observed in at least 25 individuals in our discovery set, after excluding any mixed or missing genotype calls. These comprised: 51,453 variants that were called as biallelic and passed quality filters (detailed in Supplementary Methods; including the requirement to lie in the core genome[52]); an additional 98 biallelic variants in the region of *PfEBL1* (which lies outside the core genome but otherwise appeared reliably callable); and an indicator

of the *PfEBA175* 'F' segment, which we called based on sequence coverage as described in Supplementary Methods and Supplementary Fig. 6. We included *PfEBL1* and *PfEBA175* variation because these genes encode known or putative receptors for *P. falciparum* during invasion of erythrocytes[15].

We concentrated on a set of human variants chosen as follows: we included the 94 autosomal variants from our previously reported list of variants with the most evidence for association with severe malaria[8], which includes confirmed associations at *HBB*, *ABO*, *ATP2B4* and the glycophorin locus. We also included three glycophorin structural variants[13], and 132 HLA alleles (62 at 2-digit and 70 at 4-digit resolution) that were imputed with reasonable accuracy (determined as having minor allele frequency > 5% and IMPUTE info at least 0.8 in at least one of the two populations in our dataset). We tested these variants against all 51,552 *P. falciparum* variants described above. We also included all common, well-imputed human variants within 2 kb of a gene determining a blood-group antigen (defined as variants within 2 kb of a gene in the HUGO blood-group antigen family[53] and having a minor allele frequency of 5% and an IMPUTE info score of at least 0.8 in at least one of the two populations in our dataset; this includes 39 autosomal genes and 4,613 variants in total). We tested these against all variants lying within 2 kb of *P. falciparum* genes previously identified as associated or involved in erythrocyte invasion[14,15] (60 genes, 1740 variants in total). In total we tested 19,830,288 distinct human-parasite variant pairs in the discovery dataset (Fig. 1a).

### Definition of regions of pairwise association

We grouped all associated variant pairs (defined as pairs $(v,w)$ having $BF(v,w) > 100$, where $BF(v,w)$ is the association test Bayes factor for the variant pair) into regions using an iterative algorithm as follows. For each associated pair $(v,w)$, we found the smallest enclosing regions $(R_v, R_w)$ such that any other associated pair either lay with $(R_v, R_w)$ or lay further than 10 kb from $(R_v, R_w)$ in the host or parasite genomes, repeating until all associated pairs were assigned to regions. For each association region pair, we then recorded the region boundaries and the lead variants (defined as the regional variant pair with the highest Bayes factor), and we identified genes intersecting the region and the gene nearest to the lead variants using the NCBI refGene[54] and PlasmoDB v44[55] gene annotations. Due to our testing a selected list of variant pairs as described above, in some cases these regions contain a single human or parasite variant. Supplementary Table 1 summarises these regions for variant pairs with BF > 1,000.

### Interpretation of association test results

We compared association test *P*-values to the expectation under the null model of no association using a quantile-quantile plot, both before and after removing comparisons with HbS (Supplementary Fig. 2; HbS is encoded by the 'A' allele at rs334, chr11:5,248,232 T -> A). A simple way to interpret individual points on the QQ-plot is to compare each *P*-value to its expected distribution under the relevant order statistic (depicted by the grey area in Supplementary Fig. 2); for the lowest *P*-value this is similar to considering a Bonferroni correction. However, we caution that thresholds determined by this approach are dependent on the set of tests carried out. A more defensible approach is to ask what *P*-value threshold is needed to generate confidence that a particular pair of variants is genuinely associated. This depends on both the prior probability of association and on the statistical power, with the relationship.

$$\text{posterior odds (association}|p < T) = \text{prior odds } \times \frac{\text{power}}{T} \qquad (3)$$

for any *P*-value threshold $T$ (Supplementary Methods)[50]. The term on the left is the odds of true association given observation of a *P*-value below the given threshold; the corresponding probability is therefore equal to one minus the positive false discovery rate[56]. Interpretation of (3) requires knowledge of both a relevant prior odds of association, and

the power, which in turn depends on the true effect size distribution and the underlying frequencies of the variants.

A similar approach conditional on the observed data (Supplementary Methods) leads to an analogous formula involving the Bayes factor instead of the power term and $T$. If the distribution of truly associated variant effect sizes is similar to the log $F(2,2)$ distribution we have used to calculate Bayes factors, and if variant pairs have approximately similar prior probability of association, then a fixed threshold on the Bayes factor would provide an approximately constant posterior probability of association.

We illustrate a possible computation as follows. The 51,552 *P. falciparum* variants represent around 20,000 1 kb regions of the *P. falciparum* genome, which might be thought of as approximately independent given LD decay rates[11]; similarly the human genome may be thought of as consisting of around 2 million approximately independent regions. If we take the view that a small number—say up to ten—of pairs of regions might be associated, this dictates prior odds on the order of 1 in 4 billion. A Bayes factor around $10^{10}$ would therefore be needed to generate substantial posterior odds of association, while a Bayes factor an order of magnitude higher would provide compelling evidence (posterior probability > 95%). In Supplementary Fig. 4 and Supplementary Methods we detail the analogous calculation applied to P-values. For large effect sizes on the order of OR ≈ 4, this suggests that *P*-values on the order of $1 \times 10^{-10}$ to $1 \times 10^{-12}$ might provide compelling evidence for association, depending on the allele frequencies, but weaker effects would require lower thresholds and would be less easily detectable.

It might be considered that the human variants and genes that we have considered here are among those with the highest prior plausibility for association with parasites, and thus the above choice of prior may be considered somewhat conservative. However, even under stronger prior odds on the order of 1 in 2 million (for example, assuming 10 associations among the variant pairs tested in our study), our results do not identify any associations additional to the HbS−*Pfsa* associations with very strong evidence. Particular variants may however be of further interest due to specific prior plausibility; in Supplementary Methods we give further details on putative associations with BF > $10^5$ and those involving known malaria-protective mutations in the human genome.

### Summarizing evidence for each variant

For each human variant $v$, we further summarised the evidence that $v$ is associated with variation in the parasite genome using the mean Bayes factor $BF_{avg}(v)$, computed as the average of the Bayes factor $BF(v,w)$ across all the parasite variants $w$ tested against v. Under the restrictive assumption that at most one parasite variant is associated with $w$, $BF_{avg}(v)$ can be interpreted as a model-averaged Bayes factor reflecting the evidence for association of $v$ with parasite variants; more generally $BF_{avg}$ provides a pragmatic way to combines evidence across all tested variants. We similarly define $BF_{avg}(w)$ for each parasite variant $w$ averaged over all human variants tested against $v$. $BF_{avg}$ is plotted for human and parasite variants in Fig. 1a.

A direct interpretation of these average Bayes factors can be carried in a similar way to the individual Bayes factors as described above. To illustrate, if as above we assume that around 10 of the 20,000 1 kb regions of the *P. falciparum* genome might be associated with human genetic variants among those tested, an average Bayes factor > 10,000 would be needed to indicate > 80% posterior odds of association; this is only achieved for the *Pfsa* variants in our data (Fig. 1a). This calculation can be adjusted as appropriate to take into account specific information about individual variants.

### Investigation of additional associations

In addition to the HbS-*Pfsa* associations, we also observed moderate evidence for association at a number of other variant pairs. These include associations between variation in the human gene *GCNT2* and *PfMSP4* with $BF = 2.8 \times 10^6$, and between HLA variation and multiple

parasite variants with BF in the range $10^5$–$10^6$ (Fig. 1a, Supplementary Table 1). A fuller description of the context of these SNPs can be found in Supplementary Methods. Our interpretation is that the statistical evidence for these associations is not sufficiently strong on its own to make these signals compelling without additional evidence.

### Assessment of possible confounding factors

To assess whether the observed association between HbS and *P. falciparum* alleles might be driven by confounding factors we conducted additional pairwise association tests as follows using HPTEST, based on directly typed HbS genotypes and working seperately in the two populations. Results are shown in Extended Data Fig. 3. First, we repeated the pairwise association test including only individuals overlapping the discovery dataset, and separately in the remaining set of 825 individuals. For discovery samples a set of population-specific principal components (PCs) reflecting human population structure were previously computed[8] and we included these as covariates (including 20 PCs in total). Second, across all 4,071 individuals with directly typed HbS data, we repeated tests including measured covariates as additional predictors. Specifically, we considered: (1) the age of individual at time of ascertainment (measured in years; range 0–12; treated as a categorical covariate), sex, reported ethnic group, and year of admission (range 1995–2010, treated as a categorical covariate); (2) technical covariates including an indicator of method of sequencing (SWGA or whole DNA), mean depth of coverage of the *P. falciparum* genome, mean insert size computed from aligned reads, and percentage of mixed calls; and (3) an indicator of the clinical form of severe malaria with which the sample was ascertained ('SM subtype'; either cerebral malaria, severe malarial anaemia, or other).

To assess the possibility that parasite population structure might impact results, we also included PCs computed in parasite populations as follows. Working in each population separately, we started with the subset of biallelic SNPs with minor allele frequency at least 1% from among the 51,552 analysed variants (50,547 SNPs in Gambia and 48,821 SNPs in Kenya respectively). We thinned variants by iteratively picking variants at random from this list and excluding all others closer than 1 kb (leaving 12,036 SNPs in Gambia and 11,902 SNPs in Kenya). We used QCTOOL to compute PCs using this list of SNPs. Several of the top PCs had elevated loadings from SNPs in specific genomic regions. This was especially noticeable in Kenya and included the widely reported extensive regions of LD around the *AAT1* and *CRT* regions on chromosomes 6 and 7, and also the HbS-associated chromosome 2 and 11 loci. We therefore also considered separate sets of PCs computed after excluding SNPs in chromosomes 6 and 7 (leaving 9,933 and 9,812 SNPs respectively), after excluding chromosomes 2 and 11 (10,521 and 10,421 SNPs respectively) or after excluding 100 kb regions centred on the lead HbS-associated SNPs (11,866 and 11,732 SNPs respectively). For each set of PCs, we repeated association tests including 20 PCs as fixed covariates.

For each subset of individuals, each HbS-associated variant and each set of covariates described above, we plotted the estimated effect size and 95% posterior interval, annotated with the total number of samples, the number carrying the non-reference allele at the given variant, and the number carrying heterozygous or homozygous HbS genotypes (Extended Data Fig. 3). Corresponding genotype counts can be found in Supplementary Table 3. To assess mixed genotypes calls, we also plotted the ratio of reads with reference and nonreference alleles at each site; this can be found in Extended Data Fig. 4.

### Interpretation in terms of causal relationships

Observing $OR \neq 1$ implies non-independence between host and parasite genotypes in individuals with severe disease, but does not determine the mechanism by which this could occur. Assuming $OR^{biting} = 1$, we show in Supplementary Methods that $OR = 1$ is equivalent to a multiplicative model in which host and parasite genotypes separately and multiplicatively affect disease risk (equations (S6) and (S7)

in Supplementary Methods). In general deviation from this model could arise in several ways, including through within-host selection, interaction effects determining disease tolerance, as well as potential effects not specific to individual variants but relating to disease diagnosis (similar to Berkson's paradox[57]). Our study provides only limited data to distinguish these possible mechanisms. For the HbS association described in main text, we note in Supplementary Methods that there is little evidence that the *Pfsa*+ variants are themselves associated with increased disease risk, and little evidence that the *Pfsa*+ variants associate with other host protective variants, suggesting that the observed interaction is specific to HbS. The appearance of the association in uncomplicated cases further suggests that the effect is not specifically related to infections manifesting as severe disease.

### Genotype inference from uncomplicated cases from Mali

We called genotypes from previously published RNA-seq data from 32 children ascertained with uncomplicated malaria from Mali[16] (NCBI BioProject PRJNA685106). We used BAM files of reads aligned to the 3D7 reference genome as generated previously[16]. All data were from 50 bp paired end RNA-seq of total RNA. For each *Pfsa* lead variant, we counted the number of reads aligning to the variant with the reference, alternate, or other allele, ignoring those with lower than 20 mapping quality. These read counts are presented in Supplementary Table 6 for the lead *Pfsa* mutations. We treated genotypes as mixed (and excluded these from association analysis in Supplementary Table 2) if > 10% of reads covering both alleles were observed; otherwise, we assigned a specific genotype. At *Pfsa3*, we also tabulated genotypes for the second-most associated SNP (chr11:1,057,437 T>C) because we noted one sample (AS15) that had different genotype between this and the chr11:1,085,035 T>A variant.

### Comparison of severe cases to human population controls

Equation (3) represents the odds ratio in severe cases as a ratio of relative risks for a given human allele *e* conditional on the parasite genotype at infection time. A closely related interpretation involves the relative risk of severe malaria with the observed (disease-time) parasite type,

$$RR(y) = \frac{P(\text{severe disease with parasite type } y|E=e)}{P(\text{severe disease with parasite type } y|E=0)} \quad (4)$$

A calculation shows that the OR in severe cases is also equal to the ratio $RR(y)/RR(0)$. The relative risk (equation (4)) can be computed by comparing the frequencies of host genotypes between disease cases with infection genotype *y* and in the whole population. In Supplementary Methods we extend this to show that $RR(y)$ can be estimated using multinomial logistic regression with population controls and cases stratified by parasite type as outcome levels, and with the human genotype and any covariates included as predictors. We apply this approach in Fig. 2 to estimate $RR(y)$, where y ranges over combined genotypes at the three *Pfsa* loci, conditional on the country of sampling.

### Assessing sequencing performance in HbS-associated regions

We assessed sequencing performance at the chr2:631,190, chr2:814,288 and chr11:1,058,035 loci by computing counts of reads aligning to each position ('coverage') and comparing this to the distribution of coverage across all biallelic sites in our dataset, treating each sample separately (Supplementary Fig. 5). In general coverage at the three sites was high; we noted especially high coverage at chr2:814,288 in sWGA sequencing data (for example, >90% of samples have coverage among the top 80% of that at biallelic variants genome-wide) but somewhat lower coverage in WGS samples at the chr11:1,058,035 locus. Variation in coverage between loci and samples is expected due to variation in DNA quantities, DNA amplification and sequencing processes, but we did not observe systematic differences in coverage between the different *Pfsa* genotypes at these loci. To further establish alignment accuracy,

we also inspected alignment metrics. Across all analysis samples, over 99% of reads at each location carried either the reference or the identified non-reference allele, and over 99% of these reads had mapping quality at least 50 (representing confident read alignment). These results suggest sequence reads provide generally accurate genotype calls at these sites.

## Assessing the distribution of between-chromosome LD
We developed a C++ program (LDBIRD) to efficiently compute LD between all pairs of *P. falciparum* variants. LDBIRD computes the frequency of each variant, and computes the correlation between genotypes at each pair of variants with sufficiently high frequency. It then generates a histogram of correlation values and reports pairs of variants with squared correlation above a specified level. We applied LDBIRD separately to *P. falciparum* data from Gambian and Kenyan severe malaria cases. We restricted attention to comparisons between biallelic variants that had frequency at least 5% in the given population and with at least 75% of samples having non-missing genotypes at both variants in the pair, after treating mixed genotype calls as missing, and output all pairs with $r^2$ at least 0.01 for further consideration. To avoid confounding of LD by the HbS association signal, we also repeated this analysis after excluding individuals that carry the HbS allele (with the latter results presented in Fig. 4 and Supplementary Table 4).

To summarise between-chromosome LD results we grouped signals into regions as follows. First, we observed that most variant pairs have $|r| < 0.15$ and hence $r^2 > 0.05$ is typically a substantially outlying degree of inter-chromosomal LD (Fig. 4). We therefore focussed on variant pairs (v1,v2) with $r^2 > 0.05$. To each such pair (v1,v2) we assigned a pair of LD regions (R1,R2) with the property that R1 and R2 capture all other nearby variants with high $r^2$. Specifically, R1 and R2 are defined as the smallest regions containing v1 and v2 respectively, such that no other variant pair within 10 kb of (R1,R2) has $r^2 > 0.05$. To compute R1 and R2, we implemented an iterative algorithm that successively expands the initial pair until no additional nearby pairs with high $r^2$ can be found. For each LD region pair we recorded the region boundaries and the most-correlated pair of variants. A full list of region pairs with $r^2 > 0.05$ is given in Supplementary Table 5; the highest LD pairs for *Pfsa* regions and for *PfCRT*–*PfAAT1* are shown in Fig. 4.

## Assessing the influence of covariates on LD estimates
To investigate whether the observed between-locus LD might arise due to population structure effects or due to other artefacts captured by measured covariates in our data, we used HPTEST to fit a logistic regression model of association with the genotypes at one *Pfsa* locus as outcome and the genotypes at a second *Pfsa* locus as predictor, repeating for each pair of *Pfsa* regions, in each population separately. We fit the model including each of a set of covariates as follows: (1) no covariates; (2) 20 parasite PCs; (3) technical covariates including an indicator of the type of sequencing and sequence depth (as in Extended Data Fig. 3); (4) year of admission, or (5) all of the above combined. For each set of covariates we compared the estimated odds ratio indicating the strength of association to the unadjusted odds ratio. In Kenya, across covariate sets, the minimum and unadjusted estimates were 128.0 and 128.0 (*Pfsa1*[+] vs *Pfsa2*[+]; minimum with no covariates), 218.0 vs 219.4 (*Pfsa1*[+] vs *Pfsa3*[+]; minimum when including technical covariates) and 40.2 vs 47.2 (*Pfsa2*[+] vs *Pfsa3*[+]; minimum when including parasite PCs). In Gambia the minimum and unadjusted estimates were 7.0 and 7.7 (*Pfsa1*[+] and *Pfsa3*[+], minimum when including parasite PCs). These results therefore suggest the observed LD is not substantially explained by population structure or other features of our sample that are captured by these covariates.

## Genotype inference from the Uganda Palo Alto isolate
To determine the genotype of the Uganda Palo Alto (FUP/H) isolate (Extended Data Figures 7-8), we downloaded Illumina sequence read data from the Sequence Read Archive (accessions SRR530503, SRR629055 and SRR629078, generated by the Broad Institute). All reads were 101 base pair paired end. We aligned the reads to the Pf3D7_v3 genome using bwa mem and inspected read pileups to determine the genotypes at HbS-associated mutations. These data clearly indicate that FUP/H carries the alternate allele at the *Pfsa1–3* lead SNPs as well as at chr:1,057,437 T>C (based on >98% of reads carrying the non-reference allele).

## Analysis of transcript expression from time course experiments
We analysed data from three previously reported experiments that measured the transcription of genes in 3D7 at different time stages following invasion of erythrocytes (accessions PRJEB2015[58] and PRJEB31535[59]). Data were processed using a similar pipeline to that described previously[16]. In brief, reads were aligned to a concatenated human GRCh38 / Pf3D7 genome using STAR v2.7.3a, informed by the Gencode v38 human and the PlasmoDB v52 Pf3D7 gene annotations. Reads aligning to Pf3D7 were then extracted and transcript abundance (TPM) was estimated using Salmon v1.5.1. Estimated TPM values for genes in the *Pfsa* regions, along with previously computed TPM values from Saelens et al.[16], are shown in Extended Data Fig. 7.

## Assessing the structure of *Pfsa* regions in available genome assemblies
We extracted 101 bp and 1001 bp flanking sequence centred at the chr2:631,190, chr2:814,288 and chr11:1,058,035 loci from the Pf3D7 reference sequence. We then used minimap2[60] to align these sequences to a previously generated set of genome assemblies from *P. falciparum* isolates and laboratory strains[38] (Supplementary Table 7), allowing for multiple possible mapping locations. Each flanking sequence aligned to a single location on the corresponding chromosome in all included genomes, with the exception that sequence flanking the chromosome 11 locus aligned to two locations in the ML01 sample. This sample was excluded from previous analysis[38] as it represents a multiple infection; we comment further on this below.

To further inspect sequence identity, we used MAFFT to generate a multiple sequence alignment (MSA) corresponding to the 1001 bp sequence centred at each locus. Four isolates (GA01 from The Gabon, SN01 from Senegal, Congo CD01 and ML01 from Mali) carry the non-reference 'A' allele at the chr11:1,058,035 SNP; two of these (GA01 and CD01) also carry the non-reference allele at the chr2:631,190 SNP and one (CD01) carries the non-reference allele at all three SNPs. However, expansion of alignments to include a 10,001 bp segment indicated that these four samples also carry a structural rearrangement at the chr11 locus. Specifically, GA01, SN01, CD01 and ML01 genomes include a ~1 kb insertion present approximately 900 bp to the right of chr11:1,058,035, and also a ~400 bp deletion approximately 2,400 bp to the left of chr11:1,058,035. To investigate this, we generated *k*-mer sharing 'dot' plots for *k* = 50 across the region (Supplementary Fig. 3, Extended Data Fig. 9), revealing a complex rearrangement carrying both deleted and duplicated segments. The duplicated sequence includes a segment (approximate coordinates 1,054,000–1,055,000 in Pf3D7) that contains the gene *SNRPF* ('small nuclear ribonucleoprotein F, putative') in the Pf3D7 reference. Inspection of breakpoints did not reveal any other predicted gene copy number changes in this region, including for *Pf3D7_1127000*.

As noted above, the chromosome 11 region aligns to a second contig in ML01 (contig chr0_142, Supplementary Table 7). This contig appears to have a different tandem duplication of a ~4 kb segment lying to the right of the associated SNP (approximately corresponding to the range 11:1,060,100–1,064,000 in Pf3D7; Supplementary Fig. 3). This segment contains a number of genes including dUTPase, which has been under investigation as a potential drug target[61]. We interpret this second contig as arising due to the multiple infection in this sample[38], and given challenges inherent in genome assembly of mixed samples it is

unclear whether this duplication represents an assembly artefact or a second genuine regional structural variant.

## Reporting summary

Further information on research design is available in the Nature Research Reporting Summary linked to this paper.

## Data availability

Sequence read data from whole DNA and SWGA sequencing of *P. falciparum* genomes (as detailed in Extended Data Fig. 1) are available from the European Nucleotide Archive (study accession ERP000190). A full list of relevant sample accessions can be found at http://www.malariagen.net/resource/32. Human genotype data used in this study have been described previously[5,8] and are available under managed-access terms from the European Genome–Phenome Archive (study accession EGAS00001001311), as detailed at https://www.malariagen.net/resource/25. A dataset of the human and *P. falciparum* genotypes for 3,346 severe cases of malaria used in our discovery scan (Fig. 1), and HbS genotypes and *P. falciparum* genotypes in the larger set of 4,071 severe cases with direct HbS typing (Fig. 2), is available from Zenodo (https://doi.org/10.5281/zenodo.4973476). Association test summary statistics from our discovery data (Fig. 1) are also available from Zenodo (https://doi.org/10.5281/zenodo.5722497). A full list of data generated by this study and associated resources can be found at http://www.malariagen.net/resource/32.

## Code availability

Source code for HPTEST and LDBIRD is available at https://code.enkre.net/qctool. A snapshot has also been deposited at Zenodo (https://doi.org/10.5281/zenodo.5685581).

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

**Acknowledgements** We thank the patients and staff of Kilifi County Hospital and the KEMRI-Wellcome Trust Research Programme, Kilifi for their help with this study, members of the Human Genetics Group in Kilifi for help with sample collection and processing, and the patients and staff at the Paediatric Department of the Royal Victoria Hospital in Banjul, Gambia for their help with the study. The human genetic data from severe cases and controls used in this study have previously been reported by the Malaria Genomic Epidemiology Network, and we thank all our colleagues who contributed to this previous work as part of MalariaGEN Consortial Project 1. A full list of consortium members is provided at https://www.malariagen.net/projects/consortial-project-1/malariagen-consortium-members. The MalariaGEN Pf6 open resource[11] was generated through the Malaria Genomic Epidemiology Network *Plasmodium falciparum* Community Project (https://www.malariagen.net/resource/26). The Malaria Genomic Epidemiology Network study of severe malaria was supported by Wellcome (https://wellcome.ac.uk/) (WT077383/Z/05/Z (MalariaGEN)) and the Bill & Melinda Gates Foundation (https://www.gatesfoundation.org/) through the Foundations of the National Institutes of Health (https://fnih.org/) (566 (MalariaGEN)) as part of the Grand Challenges in Global Health Initiative. The Resource Centre for Genomic Epidemiology of Malaria is supported by Wellcome (090770/Z/09/Z and 204911/Z/16/Z (MalariaGEN)). This research was supported by the Medical Research Council (https://mrc.ukri.org/) (G0600718, G0600230 and MR/M006212/1 (MalariaGEN)). Wellcome also provides core awards to the Wellcome Centre for Human Genetics (203141/Z/16/Z (WCHG)) and the Wellcome Sanger Institute (206194 (WSI)). Genome sequencing was carried out at the Wellcome Sanger Institute and we thank the staff of the Wellcome Sanger Institute Sample Logistics, Sequencing, and Informatics facilities for their contribution. T.N.W. is supported through a Senior Fellowship from Wellcome (202800/Z/16/Z). This paper is published with permission from the Director of the Kenya Medical Research Institute (KEMRI). Sequencing and initial analysis of the Mali uncomplicated malaria cases was funded by NIAID (R21AI125988 to S.M.T.). J.W.S. received support from the National Center for Advancing Translational Sciences (UL1TR002553). This research was funded in whole or in part by Wellcome as detailed above. The funders had no role in study design, data collection and analysis, decision to publish, or preparation of the manuscript. We thank D. Goldberg (Washington University in St.Louis) and M. Lee (Wellcome Sanger Institute) for their assistance with revisions.

**Author contributions** Conceptualization: G.B., E.M.L., T.N.W., K.A.R. and D.P.K. Data curation: G.B., E.M.L., T.N., M.J., C.M.N., R.D.P., R.A. and K.A.R. Formal analysis: G.B., E.M.L. and K.A.R. Funding acquisition: D.P.K. Investigation: C.H., A.E.J., K.R., E.D. and K.A.R. Methodology: G.B., K.A.R. and D.P.K. Project administration: S.G., E.D., K.A.R. and D.P.K. Resources: S.G., E.D., J.S., C.V.A., R.A., R.D.P., M.J., F.S.-J., K.A.B., G.S., C.M.N., A.W.M., N.P., J.W.S., M.D., S.M.T., C.H., A.E.J., K.R., E.D. and K.A.R. Software and visualisation: G.B. Supervision: D.J.C., U.d.A., K.M., T.N.W., S.G., K.A.R. and D.P.K. Writing: G.B., E.M.L., S.M.T., D.J.C., T.N.W., K.A.R. and D.P.K. in collaboration with all authors.

**Competing interests** The authors declare no competing interests.

**Additional information**
**Correspondence and requests for materials** should be addressed to Gavin Band, Kirk A. Rockett or Dominic P. Kwiatkowski.

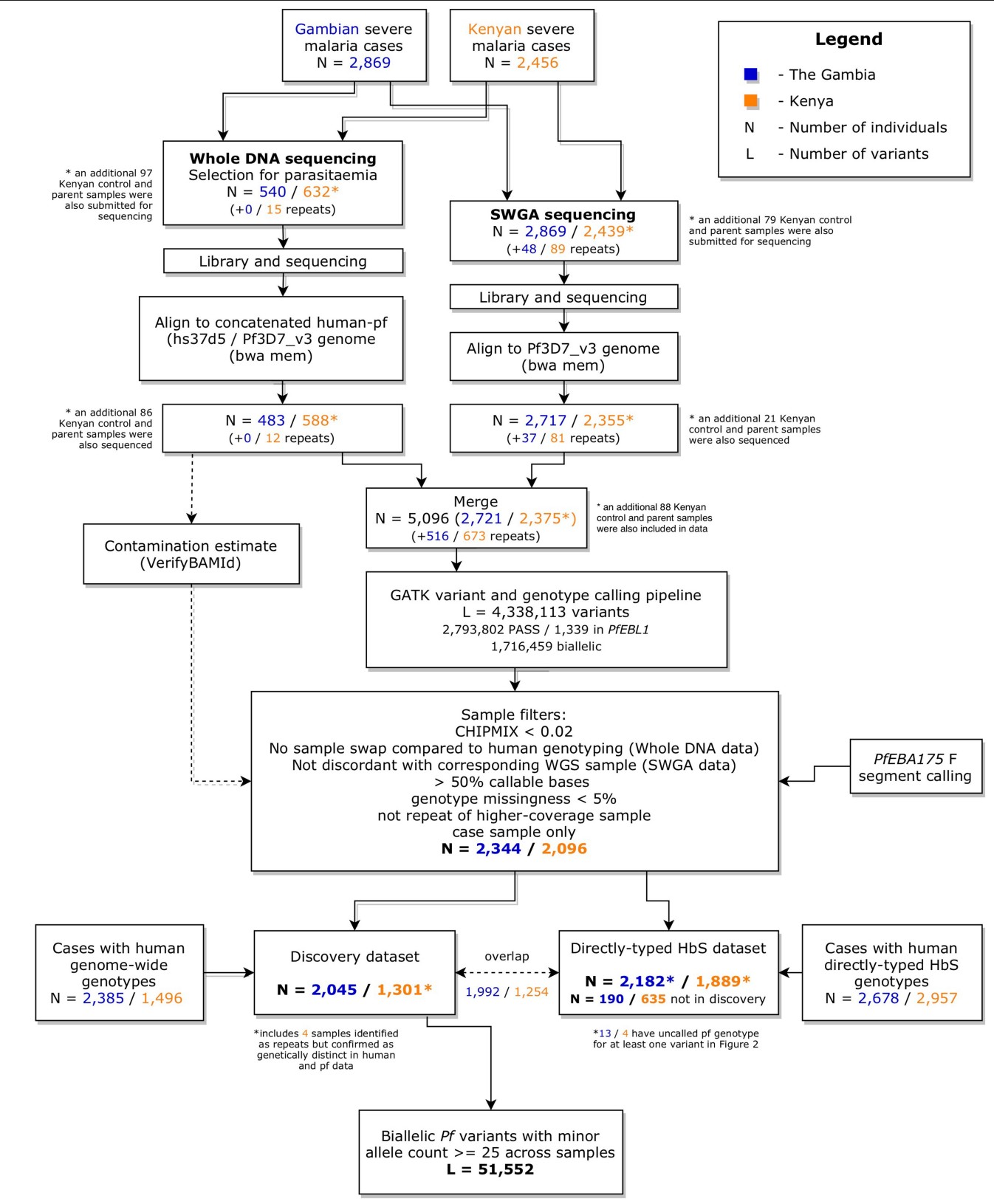

**Extended Data Fig. 1 | Flowchart showing generation and processing of *P. falciparum* (*Pf*) sequence data from 5,096 severe malaria cases.** Flowchart shows sample processing from initial selection for whole DNA and Selective Whole genome Amplification (SWGA) pipelines (top) to the curated analysis datasets (bottom). Numbers in each box show counts of severe cases in The Gambia (blue) and Kenya (orange) with the number of individuals sequenced multiple times indicated in brackets. Following curation and QC of data (large box), available *Pf* data was intersected with two existing human genotype datasets to for the analyses described in main text. The combined pf/human imputed dataset has 3,346 samples and the combined pf/human direct typing dataset contains 4,071 individuals. These two datasets have substantial overlap; 825 individuals were represented in the directly-typed data but not the imputed data and were used for replication.

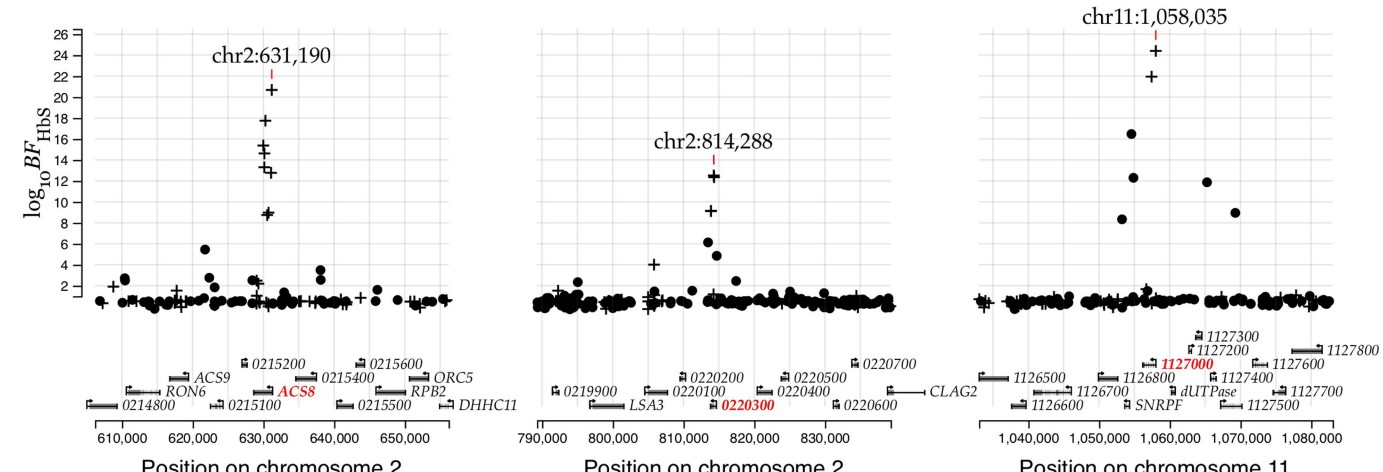

**Extended Data Fig. 2 | Evidence for association with HbS in three regions of the *Pf* genome using directly-typed HbS genotypes.** Points show evidence for association with HbS (log₁₀ Bayes Factor for test in N = 4,071 samples, y axis) based on direct typing of HbS for variants in the *Pfsa1*, *Pfsa2* and *Pfsa3* regions of the *Pf* genome (panels). Variants which alter protein coding sequence are denoted by plusses, while other variants are denoted by circles. Results are computed by logistic regression including an indicator of country as a covariate and assuming an additive model of association; missing and mixed Pf genotype calls were excluded. A corresponding plot using imputed HbS genotypes can be found in Fig. 1. The variant with the strongest association in each region is annotated and the panels show regions of length 50kb centred at this variant. Below, regional genes are annotated, with gene symbols given where the gene has an ascribed name in the PlasmoDB annotation (after removing 'PF3D7_' from the name where relevant); the three genes containing the most-associated variants are shown in red.

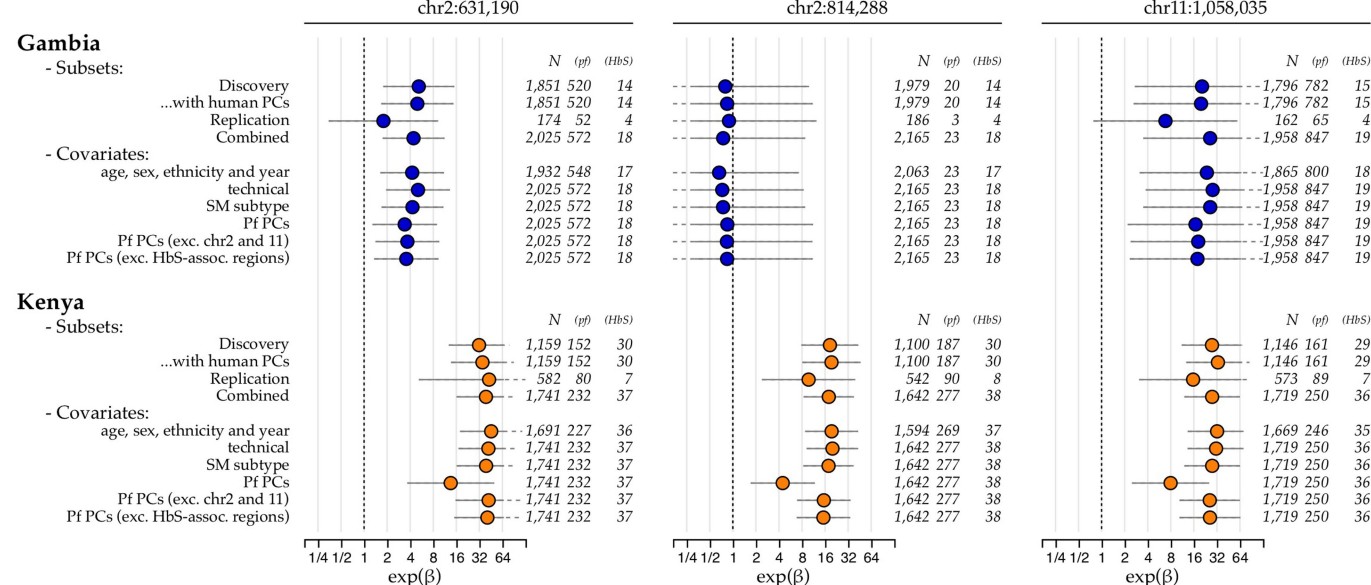

**Extended Data Fig. 3 | Odds ratios for association of HbS with the *Pfsa* variants in severe malaria cases.** Plot shows parameter estimates (points) and 95% posterior credible intervals (horizontal line segments) for the association of HbS with *Pf* genotype at each of the three *Pfsa* lead variants (columns), using several combinations of sample subsets and covariates (rows) in The Gambia and Kenya. Estimates are computed separately for each SNP using logistic regression with the given covariates included as fixed-effect terms, and are based on directly-typed HbS genotypes assuming a dominance model of HbS on *Pf* genotype. Samples with mixed *Pf* genotype calls are excluded from the regression. All estimates are made using a weakly-informative log-F*(2,2)* prior (Supplementary Methods) on the genetic effect; a diffuse log-F*(0.08,0.08)* prior is also applied to covariate effects. Row names are as follows: "Discovery": samples with human genome-wide imputed data that were included in our initial scan (Fig. 1); "Replication": the 825 additional samples that are not closely related to discovery samples (as determined previously[8]); "Combined": all samples with direct typing (as in Fig. 2 and Extended Data Fig. 2); "technical": indicators of sequencing performance including indicator of SWGA or whole DNA sequencing method for the sample, sequence read depth, insert size, and proportion of mixed genotype calls; "SM subtype": indicator of clinical presentation (cerebral malaria, severe malarial anaemia or other severe malaria) the individual was ascertained with; "Pf PCs": principal components (PCs) computed using all called biallelic SNPs having minor allele frequency at least 1% in each population and thinned to exclude variants closer than 1kb; additional rows are shown for PCs computed after excluding SNPs in chromosomes 2 and 11, or from the three regions of association shown in Fig. 1 plus a 25kb margin. Numbers to the right of each estimate show the total regression sample size, the number of samples having the non-reference allele at the given *Pf* SNP, and the number heterozygous or homozygous for HbS.

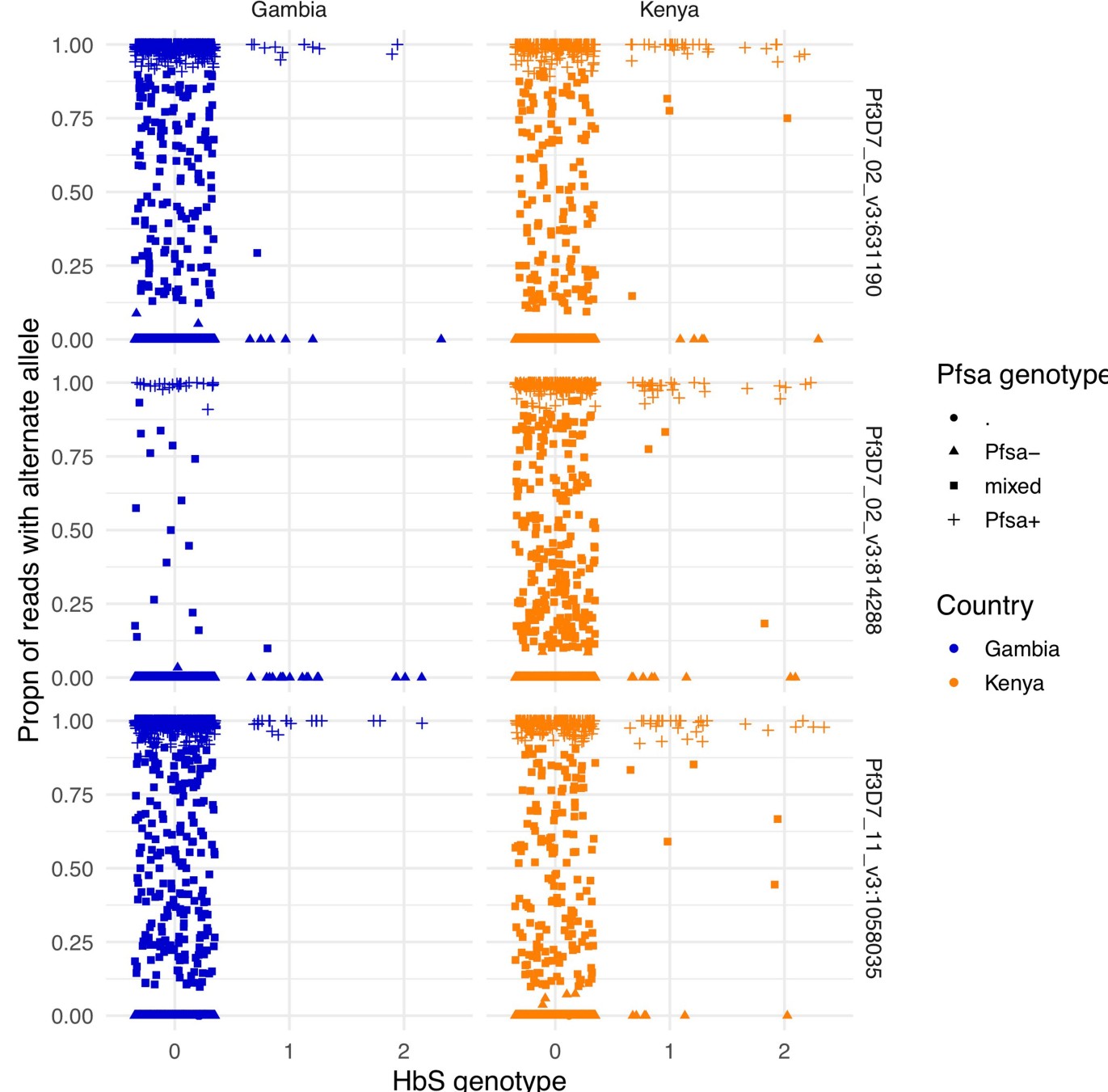

**Extended Data Fig. 4 | Allele read ratio versus HbS genotype at the three HbS-associated loci.** For each sample (points) and each of the three HbS-associated loci (rows), the figure shows the proportion of sequencing reads that carry the nonreference allele (y axis). Points are separated by country (columns) and HbS genotype (x axis); the x axis values are jittered to visually separate. The called *Pf* genotype of each sample is indicated by the shape, with mixed calls indicated by squares.

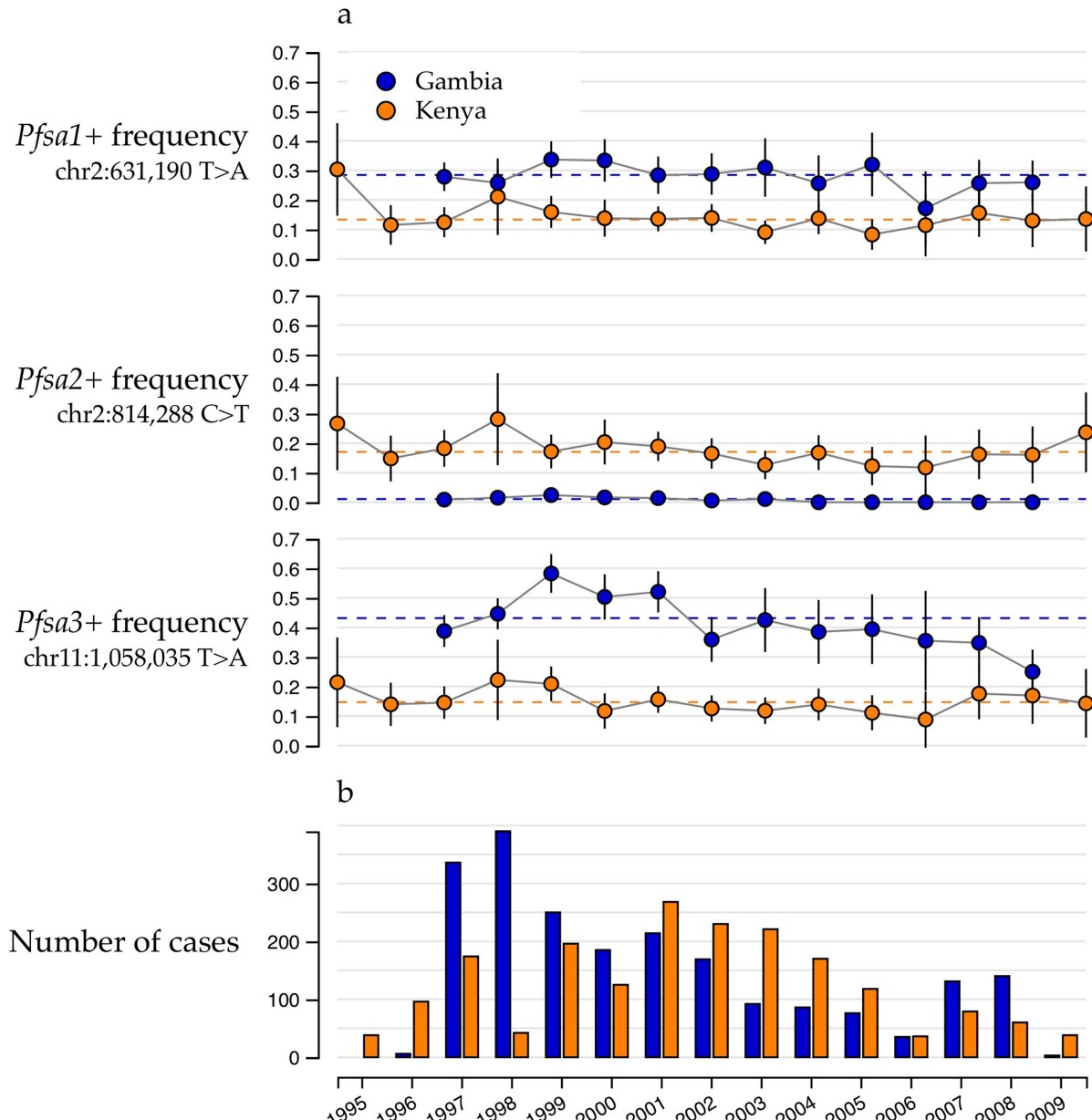

**Extended Data Fig. 5 | *Pfsa*+ allele frequency and sample size by year of ascertainment. a)** points show the sample allele frequency (y axis) for each *Pfsa* variant (rows) in severe malaria cases by year of ascertainment (x axis) and country (colour). Vertical line segments show the 95% confidence interval corresponding to each estimate. Horizontal dashed lines show the overall estimate across all years in our data, as in Fig. 3a. **b)** Bars show the total number of severe case samples in our dataset (y axis) in each country (colour) by year of ascertainment (x axis).

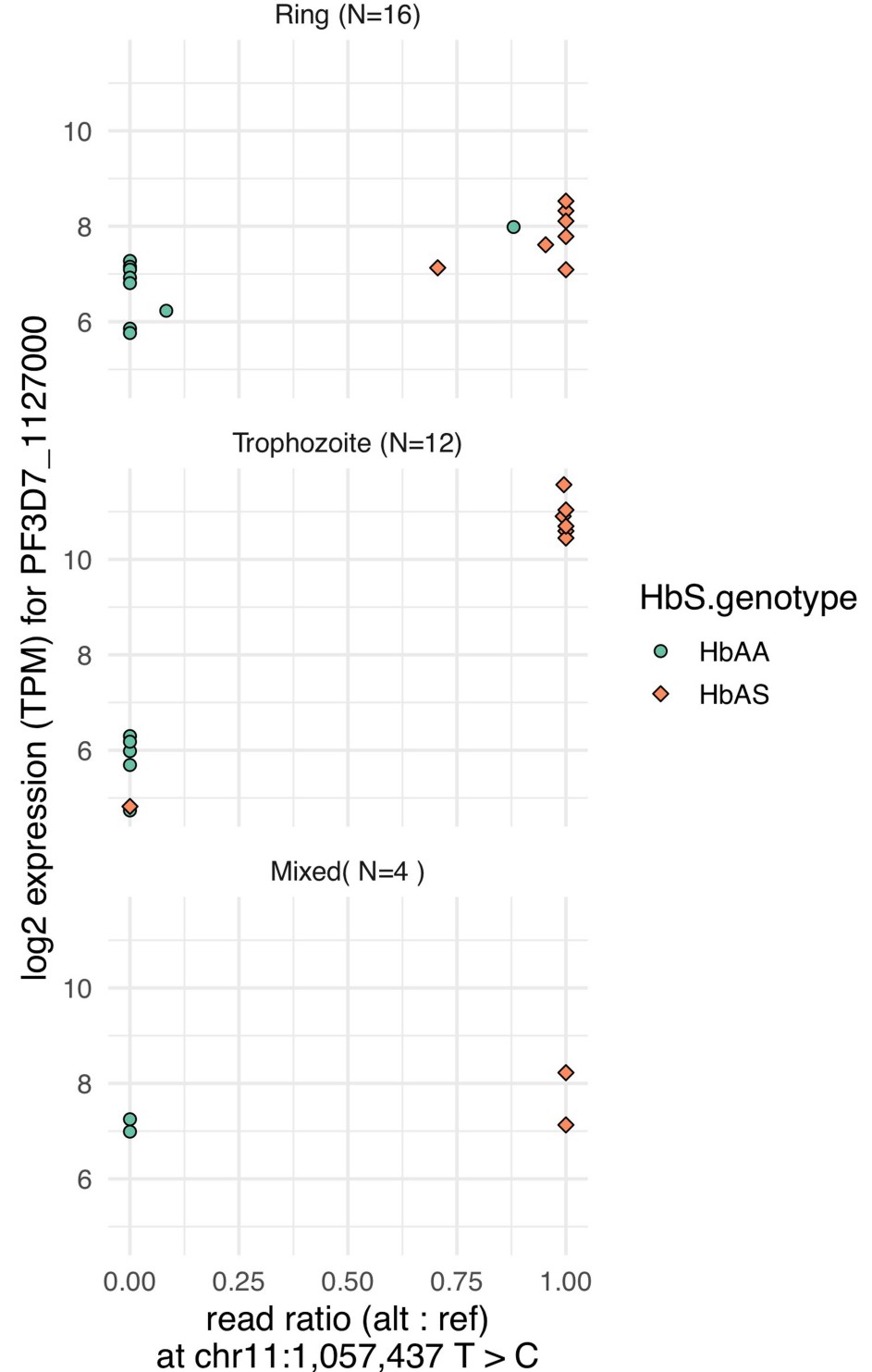

**Extended Data Fig. 6 | *Pfsa3*+ genotypes are correlated with *PF3D7_1127000* transcript levels in trophozoite-stage infections.** Plot shows the ratio of RNA-seq reads carrying the non-reference allele at the chr11:1,057,437 T > C mutation (x axis) against the estimated transcript abundance of *PF3D7_1127000* (log2 TPM, y axis), for 32 children from Mali ascertained with HbAA or HbAS genotype (colours, as detailed in the legend). Underlying data is as published by Saelens et al[16] and is further detailed in Supplementary Table 6. The plot is separated by parasite stage (panel labels) as previously inferred[16].

Among trophozoite-stage infections, we noted one infection of an HbAS individual (AS08) that has low expression of *PF3D7_1127000* (TPM = 28.3); this sample appears to have *Pfsa3*- genotype although we caution that only two reads with reasonable mapping quality were observed. Conversely, one ring-stage infection of an HbAA genotype individual (AA01) has relatively high expression (TPM = 253.2) of *PF3D7_1127000*; this sample is likely mixed as it appears to express gene copies with both *Pfsa3*- and *Pfsa3*+ genotypes, with *Pfsa3*+ predominant.

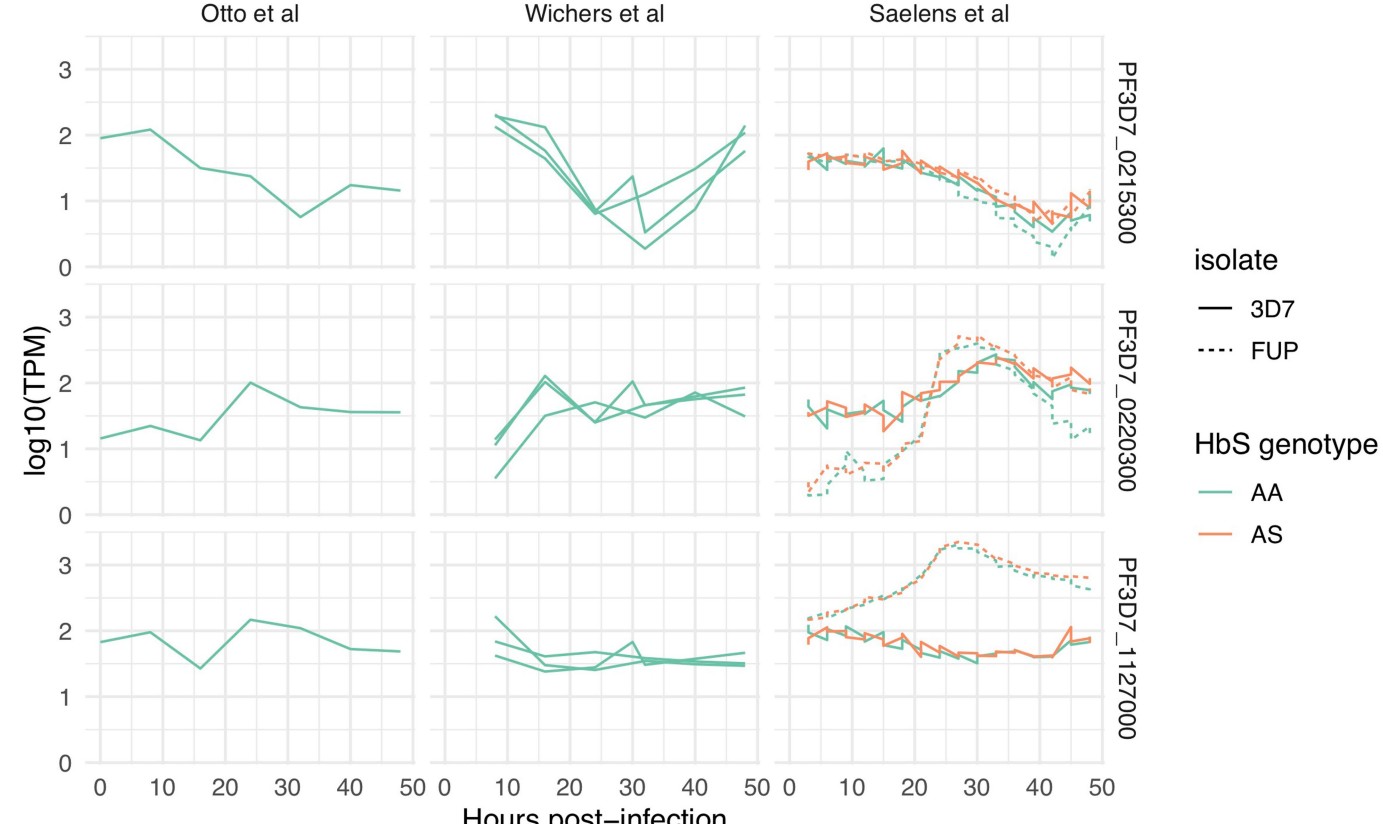

**Extended Data Fig. 7 | Estimated abundance of *Pfsa* region gene transcripts from *in vitro* intraerythrocytic time course experiments.** Plot shows the estimated relative transcript abundance (log₁₀ TPM, y axis) against hours post-infection of erythrocytes (x axis) for the three *Pfsa* region genes containing nonsynonymous sickle-associated polymorphisms (Supplementary Table 1). Data is from three studies which analysed the 3D7 isolate (Otto et al[58], Wichers et al[59] and Saelens et al[16]) as indicated by columns; the Saelens et al study also analysed the FUP/H isolate (dashed lines). Time points can be roughly interpreted[58] as: ring stage (~0-16h post-invasion); trophozoite stage (16-40h post-invasion); schizont stage (40-48h post-invasion). Replicate experiments are indicated by multiple lines in each panel; colours indicate the HbS genotype of the erythrocytes used as noted in the legend. TPM values are estimated based on reads aligning to the 3D7 reference genome; for the Saelens et al study these were used as reported previously while for the other studies we recomputed TPM as described in **Methods**.

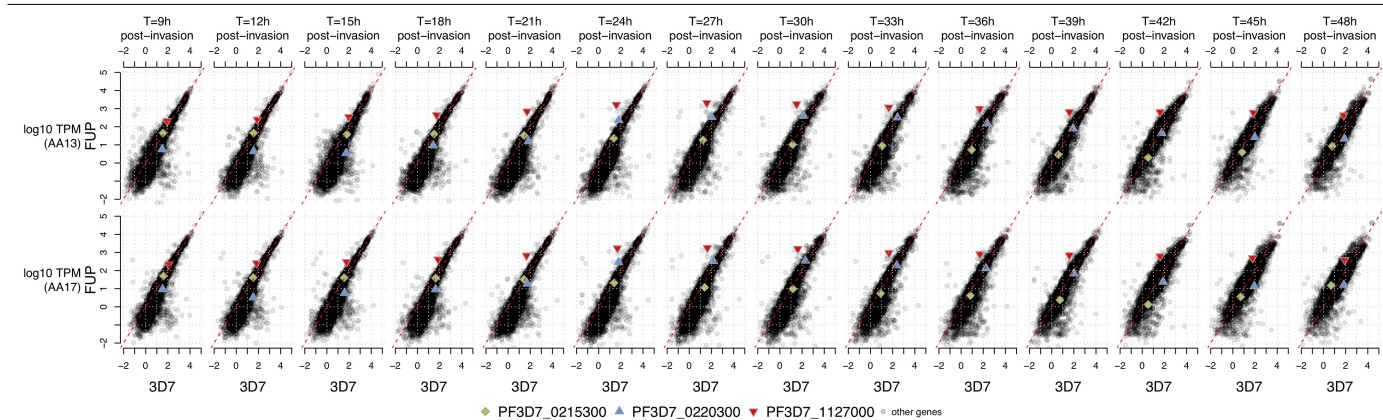

**Extended Data Fig. 8 | Estimated abundance of all transcripts in 3D7 and FUP/H parasites across the intraerythrocytic time course.** Plot shows the estimated relative transcript abundance (TPM) of *P. falciparum* genes measured in 3D7 (x axis) and in FUP/H (Uganda Palo Alto, y axis) parasites, using the data reported by Saelens et al[16]. Transcript abundance is measured *in vitro* using erythrocytes from two HbAA genotype individuals (rows), and at multiple time points post-infection (columns). TPM is measured by alignment to the 3D7 genome followed transcript quantification as described by Saelens

et al[16]. The genes *PF3D7_0215300* (*PfACS8*, *Pfsa1* locus), *PF3D7_0220300* (*Pfsa2* locus), and *PF3D7_1127000* (*Pfsa3* locus) are denoted by coloured points as shown in the legend. Both *PF3D7_1127000* and to a lesser extent *PF3D7_0220300* show an increase in expression at trophozoite stage in FUP/H parasites. We determined the genotypes of FUP/H. We determined the *Pfsa* genotypes of FUP/H as +++ (using the notation of Fig. 2) by aligning available short-read sequencing reads (SRA accessions SRR530503, SRR629055, and SRR629078; Broad Institute 2014).

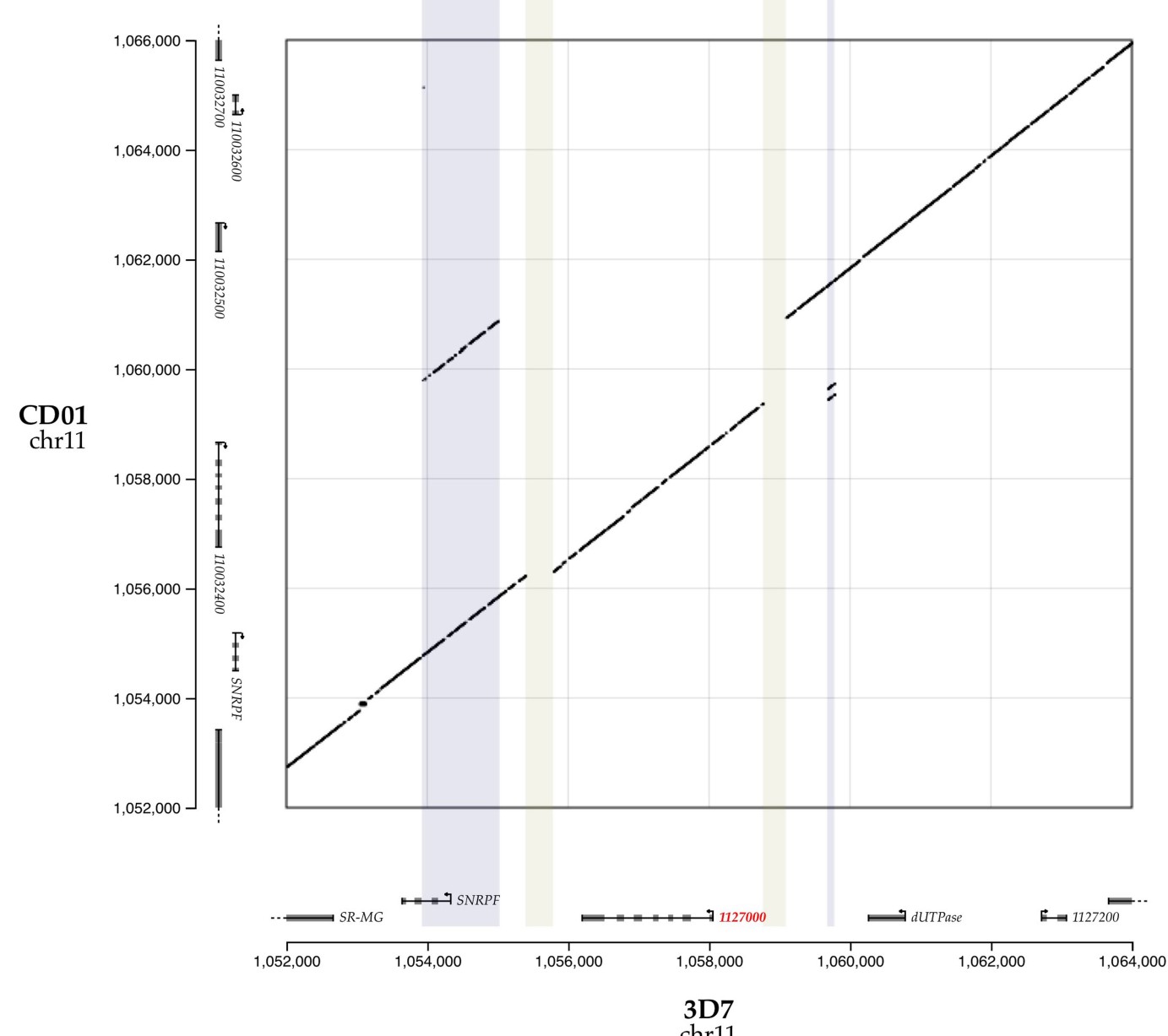

**Extended Data Fig. 9 | Structural variation at the *Pfsa3* locus.** Plot shows all DNA segments of length 50 (50-mers) that are shared identically between the 3D7 genome assembly (x axis) and CD01 genome assembly[38] (y axis) in the *Pfsa3* region. Points near the diagonal indicate similar structure, while sequences of off-diagonal points indicate structural differences between genomes. Coloured regions indicate approximate regions of 3D7 that contain increased copy number (light blue) or deletions (light yellow) in CD01 relative to 3D7.

Segment endpoints are determined by inspection of shared kmer locations and are: 1,053,925 - 1055024 (duplication); 1,055,395-1,055,784 (deletion); 1,058,765-1,059,087 (deletion); 1,059,675-1,059,777 (triplication). The CD01 assembly carries *Pfsa1+*, *Pfsa2+* and *Pfsa3+* alleles. Comparisons of 3D7 to other available assembled *Pf* genomes in *Pfsa* regions can be found in Supplementary Fig. 3.

Kirk A. Rockett
Dominic P. Kwiatkowski

# Reporting Summary

Nature Research wishes to improve the reproducibility of the work that we publish. This form provides structure for consistency and transparency in reporting. For further information on Nature Research policies, see our Editorial Policies and the Editorial Policy Checklist.

## Statistics

For all statistical analyses, confirm that the following items are present in the figure legend, table legend, main text, or Methods section.

| n/a | Confirmed | |
|---|---|---|
| ☐ | ☒ | The exact sample size ($n$) for each experimental group/condition, given as a discrete number and unit of measurement |
| ☒ | ☐ | A statement on whether measurements were taken from distinct samples or whether the same sample was measured repeatedly |
| ☐ | ☒ | The statistical test(s) used AND whether they are one- or two-sided<br>*Only common tests should be described solely by name; describe more complex techniques in the Methods section.* |
| ☐ | ☒ | A description of all covariates tested |
| ☐ | ☒ | A description of any assumptions or corrections, such as tests of normality and adjustment for multiple comparisons |
| ☐ | ☒ | A full description of the statistical parameters including central tendency (e.g. means) or other basic estimates (e.g. regression coefficient) AND variation (e.g. standard deviation) or associated estimates of uncertainty (e.g. confidence intervals) |
| ☐ | ☒ | For null hypothesis testing, the test statistic (e.g. $F$, $t$, $r$) with confidence intervals, effect sizes, degrees of freedom and $P$ value noted<br>*Give P values as exact values whenever suitable.* |
| ☐ | ☒ | For Bayesian analysis, information on the choice of priors and Markov chain Monte Carlo settings |
| ☒ | ☐ | For hierarchical and complex designs, identification of the appropriate level for tests and full reporting of outcomes |
| ☐ | ☒ | Estimates of effect sizes (e.g. Cohen's $d$, Pearson's $r$), indicating how they were calculated |

*Our web collection on statistics for biologists contains articles on many of the points above.*

## Software and code

Policy information about availability of computer code

| | |
|---|---|
| Data collection | No software was used to collect the data used in this study. |
| Data analysis | The following software was used to analyse data in this study:<br>GATK v3.6.0 (http://gatk.broadinstitute.org)<br>PICARD v2.6.0 (https://broadinstitute.github.io/picard/)<br>SNAKEMAKE v5.11.1 (https://snakemake.readthedocs.io)<br>R version 3.4.3 (https://cran.r-project.org)<br>datasette version 0.39 (https://docs.datasette.io/en/stable/)<br>d3.js version 5.15.1 (https://d3js.org)<br>QCTOOL v2.0.8 (https://www.well.ox.ac.uk/~gav/qctool)<br>LDBIRD v2.1.9 (https://code.enkre.net/qctool)<br>HPTEST v2.1.9 (https://code.enkre.net/qctool)<br>STAR v2.7.3a (https://github.com/alexdobin/STAR)<br>Salmon v1.5.1 (https://combine-lab.github.io/salmon/) |

For manuscripts utilizing custom algorithms or software that are central to the research but not yet described in published literature, software must be made available to editors and reviewers. We strongly encourage code deposition in a community repository (e.g. GitHub). See the Nature Research guidelines for submitting code & software for further information.

## Data

Policy information about availability of data

All manuscripts must include a data availability statement. This statement should provide the following information, where applicable:
- Accession codes, unique identifiers, or web links for publicly available datasets
- A list of figures that have associated raw data
- A description of any restrictions on data availability

Sequence read data from Whole DNA and SWGA sequencing of P.falciparum genomes (as detailed in Supplementary Figure 1) is available under open-access terms from the European Nucleotide Archive (study accession ERP000190). A full list of relevant sample accessions can be found at http://www.malariagen.net/resource/32. Human genotype data used in this study has been described previously[2,5] and is available under managed-access terms from the European Genome-Phenome Archive (study accession EGAS00001001311), as detailed at https://www.malariagen.net/resource/25. A dataset of the human and Pf genotypes for 3,346 severe malaria cases used in our discovery scan (Figure 1), and HbS genotypes and Pf genotypes in the larger set of 4,071 severe cases with direct HbS typing (Figure 2) is available from Zenodo under open-access terms (doi:10.5281/zenodo.4973476). A full list of data generated by this study and associated resources can be found at http://www.malariagen.net/resource/32.

# Field-specific reporting

Please select the one below that is the best fit for your research. If you are not sure, read the appropriate sections before making your selection.

☒ Life sciences  ☐ Behavioural & social sciences  ☐ Ecological, evolutionary & environmental sciences

For a reference copy of the document with all sections, see nature.com/documents/nr-reporting-summary-flat.pdf

# Life sciences study design

All studies must disclose on these points even when the disclosure is negative.

| | |
|---|---|
| Sample size | 4,171 |
| Data exclusions | Data were excluded based on quality control of sequence read data as described in Supplementary Methods. |
| Replication | The main findings were replicated in additional samples from the same populations, and in uncomplicated infections from Mali as described in Main Text. |
| Randomization | Sample genotypes were not known at time of sample ascertainment. No formal randomisation process was followed. |
| Blinding | Sample genotypes were not known at time of sample ascertainment. No formal blinding process was followed. |

# Reporting for specific materials, systems and methods

We require information from authors about some types of materials, experimental systems and methods used in many studies. Here, indicate whether each material, system or method listed is relevant to your study. If you are not sure if a list item applies to your research, read the appropriate section before selecting a response.

### Materials & experimental systems

| n/a | Involved in the study |
|---|---|
| ☒ | ☐ Antibodies |
| ☒ | ☐ Eukaryotic cell lines |
| ☒ | ☐ Palaeontology and archaeology |
| ☒ | ☐ Animals and other organisms |
| ☐ | ☒ Human research participants |
| ☒ | ☐ Clinical data |
| ☒ | ☐ Dual use research of concern |

### Methods

| n/a | Involved in the study |
|---|---|
| ☒ | ☐ ChIP-seq |
| ☒ | ☐ Flow cytometry |
| ☒ | ☐ MRI-based neuroimaging |

## Human research participants

Policy information about studies involving human research participants

| | |
|---|---|
| Population characteristics | A full description of epidemiological characteristics of participants has been published previously (https://doi.org/10.1038/ng.3107) |
| Recruitment | Participants were recruited on attendance at district hospitals in Fajara and Banjul, The Gambia, and in Kilifi, Kenya as described previously (https://doi.org/10.1038/ng.3107) |

Ethics oversight

Sample collection and design of our case-control study was approved by Oxford University Tropical Research Ethics committee (OXTREC), Oxford, United Kingdom (OXTREC 020-006). Local approving bodies were the MRC/Gambia Government Ethics Committee (SCC 1029v2 and SCC670/630) and the KEMRI Research Ethics Committee (SCC1192).

Note that full information on the approval of the study protocol must also be provided in the manuscript.

