## [Peer Review File · Nature]

Manuscript Title: Malaria protection due to sickle haemoglobin depends on parasite genotype

Reviewer Comments & Author Rebuttals

Reviewer Reports on the Initial Version:

Referee #1 (Remarks to the Author):

Previous genetic analyses have found associations between severe malaria and the sickle cell genotype (in the host; HbS), and, separately, various loci in the pathogen. Additionally, there is evidence in both the host and parasite genome of a long co-evolutionary history between humans and malaria. This raises the question of whether the parasite has developed genetic mechanisms to overcome the protective effect of the HbS genotype, and more broadly whether there are significant genetic interaction effects between the host and parasite genomes. These are important questions in a significant disease, with implications for evolution, pathogenesis and possibly treatment.

In this study, the authors use a host-pathogen interaction analysis to answer these questions. They use 3346 samples taken from cases of severe malaria in two countries. The main difficulty this study design usually encounters is a very large number of interactions weighing against a small number of samples (which are difficult to obtain, and getting independent replication cohorts is particularly challenging). The authors largely get around these issues by choosing only candidate host loci to test (though, more on this in my comments below), and using an impressive network to collect appropriate matched samples from multiple countries.

Their key finding is that three loci in the parasite (named Pfsa by the authors) are associated with increased incidence of severe malaria in HbS-positive hosts. Four lines of evidence are presented to support this conclusion:

- 1) Biological relevance of the host genotype involved.
- 2) A GWAS-type interaction analysis of candidate host loci and all pathogen loci, which finds a significant interaction between HbS and Pfsa.
- 3) Population-level association between HbS and Pfsa frequencies in the MalariaGen host populations/countries.
- 4) Strong LD between the pathogen loci, despite not being in physical linkage.

I found these approaches to be complementary, and taken together also found them to be convincing evidence for the result presented. The analysis was thorough and methodical, and the conclusions and importance not to be overstated. From a statistical genetics perspective, this study is an excellent example of how to do an interaction analysis well, and provides detailed methods and software to do so. From a disease perspective, I think this is a genuinely interesting finding which will inspire further research.

My overall opinion of this paper is very positive. I do still have some specific comments which I think could improve the paper's presentation, and help strengthen some of the evidence presented.

Major comments

- 1) On the selection of host loci to test. The authors simply state which host loci they used in their interaction analysis, but do not provide any rationale as to why all of the rest of genotyped loci in the host have been discarded (which is mentioned as being available in the preceding sentence). Presumably the reason is either to avoid a stricter multiple testing threshold, or due to computational difficulties? I think this analysis should be included: given this high-quality dataset

it would be useful hypothesis-generation for other studies, and even if it does not provide significant findings in these samples, will be important data for future studies and meta-analysis. Relatedly, the authors clearly prefer a Bayesian approach to significance testing. This is fine (it is carefully explained and justified, and I appreciated the inclusion of frequentist tests for readers more familiar with that approach). However, in selecting candidate host loci they have essentially applied a strict prior of zero to the rest of the host genome, without justification. In this framework, I think it would instead be preferable to analyse all of the variation, but give a strong prior of association to candidate loci, and a weaker prior to all other host loci.

2) The authors use logistic regression for their association, which they correct for host and pathogen population structure with PCs. The host samples have been picked to have limited population structure, and the authors present a QQ-plot to show adequate control. However, in my experience of pathogen GWAS this method of adjustment is often insufficient, and isn't always reflected well in the inflation value of the QQ-plot. This is because of 'stronger' pathogen population structure not captured by use of a few PCs (see Earle et al Nat Micro 2016). It would reassure me on this point if the authors were able to use a linear mixed model approach to confirm their results, ideally with both host and pathogen kinship matrices (possible in e.g. fastlmm), at least for the leading association. This typically accounts for stronger population structure more accurately. Removing the Pfsa loci from the kinship matrices, or using the leave-one-chromosome-out method, would be useful replication of the results changing when including/excluding the Pfsa loci from the PC calculation.

3) The approach used as evidence for possible co-selection between the Pfsa loci used a measure of LD. This is unable to account for population structure, and I would encourage the authors to consider an approach based on mutual information (such as SpydrPick) which can account for relatedness.

More importantly, I didn't think these results were clearly presented by figure 4 (table 1 is good). The main things I think are missing are physical/chromosome distance, and clear identification of the Pfsa loci. I would encourage the authors to investigate a different presentation of these results, rather than the distributions of r . I would suggest a full epistasis/co-selection analysis combined with a circos plot or similar (e.g. fig 2 in Skwark et al 2017 doi:10.1371/journal.pgen.1006508), but I'm sure other presentations are possible.

4) The distinction between analysis of multiple populations/countries, and 'replication' cohorts (which I believe only include candidate host loci genotyped via sequenom?) were a little unclear. It would be useful to clarify the distinction between replication cohorts and host populations (I found fig S1 helpful, maybe this could be described more in the main text). Or perhaps note the 'replication' split by genotyping method is essentially arbitrary, and replication between the Gambia and Kenya is likely more meaningful.

Figure 2 compounds this problem. It splits the Gambia and Kenya which is useful, but also was at odds with what I understood by 'replication' by this point in the text. A forest plot by cohort would have been better: to this end I found that figure S7 cleared up a lot of my questions and misunderstandings on what was tested and found in each cohort. Perhaps an edited version of fig S7 could be switched with figure 2?

Minor comments:

- In my opinion, Figure 1 and Figure S4 should be switched. Stratifying by host genotype and showing just the pathogen genotype is somewhat subtle to the general reader, and hides the novel interaction approach of this paper.

- The authors use a headline figure of 4171 samples in the abstract, and start the results with a number of 5096, but really were only able to analyse 3346 of these with host genotypes. It would be more honest to state 3346 as the number actually used.

- I think that a simple power analysis for host-pathogen interaction analysis would be very useful for designing future studies of malaria host-pathogen interactions, given the authors are now able to estimate realistic effect sizes.

- None of the three previous host-pathogen interaction analyses I am aware of have been cited, and probably should be:

Bartha, I. et al. A genome-to-genome analysis of associations between human genetic variation, HIV-1 sequence diversity, and viral control. *Elife* 2, 1–16 (2013)

Azim Ansari, M. et al. Genome-to-genome analysis highlights the effect of the human innate and adaptive immune systems on the hepatitis C virus. *Nat. Genet.* (2017) doi:10.1038/ng.3835

Lees, J. A. et al. Joint sequencing of human and pathogen genomes reveals the genetics of pneumococcal meningitis. *Nat. Commun.* 10, 2176 (2019) [Noting CoI that this is my own paper]

I think from a methodological perspective, contrasting this study with these others would be interesting to statistical genetics readers.

- Could the authors comment on why Pfsa2+ is so rarely found in the Gambia, despite apparent co-selection in Kenya?

- Could the low, but constant, frequencies of the alleles be explained by some form of frequency-dependent selection on the parasite?

- The linked resource page <https://www.malariagen.net/resource/32> is empty. MalariaGen are clearly good at sharing information, but I would ask this to be filled out before publication if possible, to assist with review.

- Did the authors consider applying the harmonic mean p-value approach, which can increase power in host-pathogen interaction analysis (Wilson PNAS 2018 doi: 10.1073/pnas.1814092116)? Or is there a reason for not using this here?

Typographical/very minor:

- Do the authors state what Pfsa stands for?

- Figure 2 shows a table in a figure, which is difficult to read and extract information from. This is done in a few places, including supplementary figures. Where possible, it would be preferable to add supplementary tables with this information.

- The authors talk about two million genotyped Pf variants, but analyse just 51225 of them. Some intuition in the main text for this factor of 40 decrease would be appreciated (mostly very rare genotypes?).

- Noting the multiple testing threshold along with the frequentist results would be useful, given this going to be more stringent than usually expected.

- In figure 3, making sure Kenya and the Gambia are labelled in all the panels would be helpful (missing in row 2, panel c).

John Lees

Referee #2 (Remarks to the Author):

This clearly presented manuscript describes an apparent association between *Plasmodium falciparum* malaria parasite genetic variants and the sickle cell haemoglobin trait (HbS), a trait

originally hypothesized to confer resistance to malaria by Tony Allison in 1954 and later experimentally validated by others. As such, this article represents a fascinating update on this story of co-evolution between humans and malaria parasites.

The evidence for this counter-adaptation by malaria parasites comes in multiple forms: 1) Statistical association between the variants in human vs. parasite genomes from sampled infections, 2) an enhanced relative risk of severe malaria in HbS individuals from parasites harboring the candidate variants (albeit in a small sample size), 3) a population-level correlation in the frequency of HbS and the Pfsa1-3+ parasite alleles, and 4) strong linkage disequilibrium (LD), even between chromosomes, for the candidate parasite loci, suggesting co-variation maintained by selection.

While any one of these lines of evidence would be insufficient to warrant confidence in this association, the combined force of all four observations elevates this hypothesis and will motivate followup studies of mechanism and population biology. This work will be of broad interest as an update to a textbook example of host/pathogen co-evolution.

I have minor questions and clarifications for the authors to address, as follows:

- 1) As the authors note, it is curious that strong LD is maintained between Pfsa1/2/3 given modest frequencies of HbS in most populations. Do they exhibit stronger LD in HbS-carriers (AS) individuals than AA individuals? A difference in the magnitude of parasite LD between these host population compartments could be used to infer the magnitude of selection during the establishment of bloodstage infections in HbS individuals.
- 2) Is there an association between Pfsa1/2/3 alleles and HbS in non-severe malaria cases? The apparent abrogation of the protective effect of HbS for severe malaria is interesting, but could the parasite alleles also be enhancing the ability of parasites to establish infections and non-severe malaria cases in HbS carriers?
- 3) Are there any hypotheses to explain why the HbS/MSP1 association was not replicated in this study? This deserves some comment.
- 4) In Figure 1 it is evident that multiple variants are associated with HbS at each parasite locus. Do Pfsa1/2/3 show evidence of recent positive selection in the parasite genome (ie long haplotypes from sweeps), or do they look like old, balanced polymorphisms? Does Pfsa2, being more limited in geographic distribution, sit on a longer haplotype?

Referee #3 (Remarks to the Author):

This is a strong paper with many interesting findings that obviously have high scientific and indeed clinical importance. The authors find good evidence that there are polymorphisms in the malaria genome that lead to more severe disease in individuals with the sickle cell genotype. These polymorphisms are more common in regions where the sickle cell genotype is more common and are in unexpectedly strong linkage disequilibrium with each other, given they are on different chromosomes.

The complicating factor in this analysis is population structure. It is quite hard to get one's head around all the possibilities, since there is human population structure and parasite population structure, which can be correlated with each other, via geography or indeed via natural selection and there can also be differences in the rate of severe or moderate malarial disease that are due to human factors that are associated with geography. One of the disease loci does contribute to an important PC, which does reduce the association signal between the locus and severe malaria, but presumably does leave the overall set of associations intact.

I would say that population structure makes interpretation of any one part of the results more difficult but that the patterns are very striking. I think the paper as it stands gets the balance right in terms of highlighting the caveats, while making the observations and their prima facie interpretations clear. The main text figures represent a good presentation of the basic underlying data. More elaborate figures are possible but I am not sure they would be beneficial.

So in sum, I do not have any major suggestions. An extremely minor one is that in Figure 2, (controls) is not properly aligned, which did actually make it a little hard for me to understand the figure when I first looked at it.

Referee #4 (Remarks to the Author):

In this manuscript, the authors estimate associations between human loci that have been associated with risk of severe malaria and parasite genotypes, assuming an additive interaction. They identify three parasite loci, in linkage disequilibrium with each other, that are significantly associated with presence of HbS, including two genes on chromosome 2 and one on chromosome 11. While almost all infections in HbS individuals contained mutations at these loci, these mutations were also common in non-HbS individuals, including individuals without severe malaria. The study generates interesting hypotheses that could be explored further, even in the context of this study, but that will require functional validation to determine mechanisms and the impact of the findings. Specific comments are listed below.

- The authors provide only one paragraph describing the premise behind the study, what is already known about the research question, and what the knowledge gaps are. This short paragraph did not convey the potential impact of the study and why they are doing it.

- It would be helpful for the authors to include what is now Supplementary Figure 1 in the main text, as it is difficult to follow the source of different samples, which underwent direct WGS versus sWGA followed by WGS, which had genome-wide human data versus genotyping, etc. In addition, including a sentence or two briefly describing the criteria for removing closely related individuals would be helpful.

- Can the authors explain why they looked at an additive interaction between loci? Often a multiplicative model is used to assess interaction terms for measures of effect like OR and RR.

- Likewise, for the analysis of the effect of HbS on severe malaria presented in Figure 2b, it might be better to construct a model including an interaction term representing the Pfsa+ loci, rather than presenting a stratified analysis with very small numbers of individuals for comparison. Also, was parasitemia included as a potential confounding variable, or was this variable used in the definition of severe malaria?

- Although I like that Figure 1 shows the context of where the associated loci are with respect to other genes in the genome, I find supplementary Figure 4 to be more comprehensive and visually appealing.

- Did the authors estimate the association between the Pfsa+ loci and severe malaria in individuals without HbS? Also, the frequency of Pfsa+ alleles seems similar between the severe cases and the population controls from MalariaGEN Pf6K (as noted by the authors). The data from Pf6K presumably primarily represent cases of uncomplicated malaria. Is it possible that these loci are simply virulence factors that increase the likelihood of clinical malaria, regardless of HbS status?

- Figure 3c represents an ecological comparison that does not represent associations at an individual level; therefore, it should be interpreted cautiously and not be used to attribute a causal

association.

- It would be helpful for the authors to delve more into the biological plausibility of the three loci identified in this study in modifying the effect of HbS on severe malaria. There is very little discussion of the proteins encoded by these genes. What are the predicted functions (GO terms), essentiality, pathways, timing of expression, etc.? How might this information support the contribution of these loci to severe malaria? Much of this information can be found in plasmDB. For example, the gene Pf3D7_0220300 has been identified as a candidate gene for virulence and encodes an exported protein. The functional studies may be beyond the scope of this paper (although such experiments would greatly strengthen the paper for publication in a high impact journal such as Nature), but currently there is very little interpretation of the results in the manuscript as written.

Author Rebuttals to Initial Comments:

Response to referees comments

In our response we have used the following typography:

- *Grey italic text with grey background* has been used to report the referees' comments. Where appropriate we have also numbered these for reference.
- Plain text without italics or highlighting is used to detail our replies to referees' comments
- **Red text in a serif font** is used to describe text that is quoted from our manuscript, with **yellow highlights** used for changed, added or updated text.

Changes are also highlighted in the revised manuscript.

Referee #1 (Remarks to the Author):

Previous genetic analyses have found associations between severe malaria and the sickle cell genotype (in the host; HbS), and, separately, various loci in the pathogen. Additionally, there is evidence in both the host and parasite genome of a long co-evolutionary history between humans and malaria. This raises the question of whether the parasite has developed genetic mechanisms to overcome the protective effect of the HbS genotype, and more broadly whether there are significant genetic interaction effects between the host and parasite genomes. These are important questions in a significant disease, with implications for evolution, pathogenesis and possibly treatment.

In this study, the authors use a host-pathogen interaction analysis to answer these questions. They use 3346 samples taken from cases of severe malaria in two countries. The main difficulty this study design usually encounters is a very large number of interactions weighing against a small number of samples (which are difficult to obtain, and getting independent replication cohorts is particularly challenging). The authors largely get around these issues by choosing only candidate host loci to test (though, more on this in my comments below), and using an impressive network to collect appropriate matched samples from multiple countries.

Their key finding is that three loci in the parasite (named Pfsa by the authors) are associated with increased incidence of severe malaria in HbS-positive hosts. Four lines of evidence are presented to support this conclusion:

- 1) Biological relevance of the host genotype involved.*
- 2) A GWAS-type interaction analysis of candidate host loci and all pathogen loci, which finds a significant interaction between HbS and Pfsa.*
- 3) Population-level association between HbS and PfsA frequencies in the MalariaGen host populations/countries.*
- 4) Strong LD between the pathogen loci, despite not being in physical linkage.*

I found these approaches to be complementary, and taken together also found them to be convincing evidence for the result presented. The analysis was thorough and methodical, and the conclusions and importance not to be overstated. From a statistical genetics perspective, this study is an excellent example of how to do an interaction analysis well, and provides detailed methods and software to do so. From a disease perspective, I think this is a genuinely interesting finding which will inspire further research.

My overall opinion of this paper is very positive. I do still have some specific comments

which I think could improve the paper's presentation, and help strengthen some of the evidence presented.

Thank you for this summary and the points raised below.

Major comments:

1.1) On the selection of host loci to test. The authors simply state which host loci they used in their interaction analysis, but do not provide any rationale as to why all of the rest of genotyped loci in the host have been discarded (which is mentioned as being available in the preceding sentence). Presumably the reason is either to avoid a stricter multiple testing threshold, or due to computational difficulties? I think this analysis should be included: given this high-quality dataset it would be useful hypothesis-generation for other studies, and even if it does not provide significant findings in these samples, will be important data for future studies and meta-analysis.

We agree that a genome-to-genome analysis is worthwhile in principle. However, we have decided not to add this to our current manuscript. The reasons for this are:

- 1) In our manuscript we deliberately aim to test a strong biological hypothesis, specifically that malaria parasites may have evolved to overcome resistance conferred by naturally occurring protective host mutations. This motivates looking for association between *Plasmodium falciparum* (*Pf*) variation and the host loci that might be plausibly involved in these interactions. We think the categories of human mutations we analyse cover most of the accessible variation that falls into this category (based on prior work on genetic aetiology of malaria infection).
- 2) Using the above approach we have discovered what is clearly an important feature of genetic susceptibility to malaria infection that is of substantial interest. Moreover, the *Pfsa* loci involved have several unusual features which add additional support and interest to our core finding. We think it is right that we focus our manuscript on describing this association, its implications and the associated features in as much detail and clarity as possible.
- 3) Adding a full genome-genome association tests for millions of pairs of additional variants, with less prior evidence, will dilute the strength of our paper. Since most variant pairs will not be genuinely associated, this scan will list many false positive associations at or near the top of the list. These results will require considerable work to make sense of, including statistical methods to aggregate signals in biologically plausible candidates, and examination of functional evidence for the signals at the top of such a scan. This work will take up space in our paper that we believe should be devoted to our key discovery. We feel that a genome-genome analysis is better suited to a full treatment in a separate study.

We have now updated our text to further clarify our rationale for choosing this set of variants:

“We used a logistic regression approach to test for pairwise association between these *P. falciparum* variants and four categories of human variants that are plausibly associated with malaria resistance: i. known autosomal protective mutations, including HbS (within *HBB*), the common mutation that determines O blood group (within *ABO*), regulatory variation associated with protection at *ATP2B4*¹⁻³ and the structural variant DUP4, which encodes the Dantu blood group phenotype⁴; ii. variants that showed suggestive but not conclusive evidence for association with severe malaria in our previous GWAS¹; iii. Human Leukocyte Antigen (HLA) alleles and additional glycophorin structural variants that we previously imputed in these samples^{1,4}; and iv. variants near genes that

encode human blood group antigens, which we tested against the subset of *P.falciparum* variants lying near genes which encode proteins important for the merozoite stage^{5,6}, as these might conceivably interact during host cell invasion by the parasite. Although several factors could confound this analysis in principle – notably, if there were incidental association between human and parasite population structure – the distribution of test statistics suggested that our test was not affected by systematic confounding after including only an indicator of country as a covariate (**Supplementary Figure 3**), and we used this approach for our main analysis. A full list of results is summarised in **Figure 1a** and **Supplementary Table 1.**”

1.1b) Relatedly, the authors clearly prefer a Bayesian approach to significance testing. This is fine (it is carefully explained and justified, and I appreciated the inclusion of frequentist tests for readers more familiar with that approach). However, in selecting candidate host loci they have essentially applied a strict prior of zero to the rest of the host genome, without justification. In this framework, I think it would instead be preferable to analyse all of the variation, but give a strong prior of association to candidate loci, and a weaker prior to all other host loci.

Please see our comments above for our rationale on this point. We think the reviewer’s suggestion makes sense in principle, but in practice we think our current approach has multiple advantages in enabling us to report our key discovery and focus on its implications.

1.2) The authors use logistic regression for their association, which they correct for host and pathogen population structure with PCs. The host samples have been picked to have limited population structure, and the authors present a QQ-plot to show adequate control. However, in my experience of pathogen GWAS this method of adjustment is often insufficient, and isn’t always reflected well in the inflation value of the QQ-plot. This is because of ‘stronger’ pathogen population structure not captured by use of a few PCs (see Earle et al Nat Micro 2016). It would reassure me on this point if the authors were able to use a linear mixed model approach to confirm their results, ideally with both host and pathogen kinship matrices (possible in e.g. fastlmm), at least for the leading association. This typically accounts for stronger population structure more accurately. Removing the *Pfsa* loci from the kinship matrices, or using the leave-one-chromosome-out method, would be useful replication of the results changing when including/excluding the *Pfsa* loci from the PC calculation.

In response to this comment we have implemented a linear mixed model association test for the HbS and *Pfsa* variants as suggested by the reviewer. Specifically, we used FaST-LMM to re-test for association including genetic relatedness matrices (GRMs) computed both from human and parasite genetic variants as random effects, using the discovery data of 3,346 samples. The results are summarised in **Table R1** below.

The results were qualitatively similar to those using including principal components presented in Supplementary Figure 7 (now renumbered as Supplementary Figure 5). Inclusion of the human GRM had little effect on association test, but inclusion of a GRM computed from genome-wide *Pf* genetic variants reduced the association signal to a limited extent. However, the association was largely restored when removing the *Pfsa* regions or the two relevant chromosomes from the GRM computation. The corresponding estimates of the contribution of the *Pf* GRM to covariance was very high (with both the overall heritability parameter and the mixing parameter for the *Pf* GRM close to 1). These results are consistent with the observations made in our paper that these variants are highly structured - and covariant in the sense of LD - within and across populations.

However, this also highlights an issue applying the linear mixed model approach to our data: we found that P-values from the LMM approach were orders of magnitude lower than those

from logistic regression applied to the same data. (For example: $P < 5 \times 10^{-18}$ at *Pfsa1* for the linear mixed model test in Kenya, compared to $P = 2 \times 10^{-11}$ we observed using a logistic regression test in **Supplementary Table 2**). We think the P-values from the LMM strongly overstate the true statistical evidence and that they arise because of the model misspecification inherent in applying a linear model (with Gaussian error distribution) to binary outcome data. To confirm this, we used the `glm()` function in R to re-fit simple logistic and linear regression models for the effect of HbS on *Pfsa1-3*, separately in each population and including no covariates, using the same genotype data as for the LMM. This showed the same behaviour, i.e. P-values from the linear model were much lower than those from logistic regression (e.g. $P=6.3 \times 10^{-25}$ in Kenya using a linear model, similar to the FaST-LMM result; compared to $P=1.5 \times 10^{-11}$ for the logistic model, similar to our discovery analysis result).

We note that LMM approaches are widely applied to control for population structure in genome-wide association test methods (including for binary phenotypes) and in many settings this misspecification is not thought to be a problem. The acute nature of the problem in our study is likely to be due to the extremely strong effect size of the HbS-*Pfsa* association (such that the logistic link is not closely modelled by a linear function across the range of predictor values).

Because of the observations made above, we have chosen not to include the LMM analysis in our revised paper, as we think the P-values are misleading. However we are happy to reconsider including this if the reviewer feels strongly that this is the right approach.

Population	Pf GRM variants	P-value	Effect size	Standard error	% variance explained	Heritability	Mixing
Pfsa1 chr2:631,190 T > A							
Gambia	all	1.48E-03	0.0192	0.0060	7.3	1.00	1.00
	Excluding Pfsa regions	3.14E-04	0.0249	0.0069	8.3	1.00	1.00
	Excluding chr 2 / 11	3.30E-04	0.0257	0.0071	8.2	1.00	1.00
	No Pf GRM	7.89E-06	0.0470	0.0105	10.2	0.09	0.00
Kenya	all	5.90E-18	0.0687	0.0078	24.6	1.00	1.00
	Excluding Pfsa regions	2.50E-21	0.0826	0.0085	26.9	1.00	1.00
	Excluding chr 2 / 11	5.13E-22	0.0856	0.0087	27.3	1.00	0.88
	No Pf GRM	3.09E-25	0.0990	0.0093	29.3	0.72	0.00
Pfsa2 chr2:814,288 C > T							
Gambia	all	6.46E-01	-0.0006	0.0014	1.0	1.00	1.00
	Excluding Pfsa regions	6.56E-01	-0.0006	0.0015	1.0	1.00	1.00
	Excluding chr 2 / 11	7.53E-01	-0.0005	0.0015	0.7	1.00	1.00
	No Pf GRM	6.63E-01	-0.0010	0.0023	1.0	0.34	0.00
Kenya	all	2.83E-11	0.0609	0.0091	19.5	1.00	1.00
	Excluding Pfsa regions	1.31E-14	0.0777	0.0099	22.6	1.00	1.00
	Excluding chr 2 / 11	4.10E-15	0.0796	0.0100	23.0	1.00	1.00
	No Pf GRM	1.20E-16	0.0915	0.0109	24.2	0.34	0.00
Pfsa3 chr11:1,058,035 T > A							
Gambia	all	4.54E-02	0.0138	0.0069	4.7	1.00	1.00
	Excluding Pfsa regions	1.96E-02	0.0176	0.0075	5.4	1.00	1.00
	Excluding chr 2 / 11	9.21E-03	0.0207	0.0079	6.1	1.00	1.00
	No Pf GRM	2.06E-04	0.0437	0.0118	8.6	0.02	0.00
Kenya	all	5.84E-18	0.0727	0.0083	24.7	1.00	1.00
	Excluding Pfsa regions	2.38E-22	0.0886	0.0089	27.7	1.00	1.00
	Excluding chr 2 / 11	1.72E-22	0.0892	0.0090	27.8	1.00	1.00
	No Pf GRM	3.83E-24	0.1017	0.0098	28.8	0.34	0.00

Table R1: linear mixed model results for the HbS-*Pfsa* association. Association test results for the association of HbS with the three *Pfsa* lead variants using the linear mixed model implemented in FaST-LMM. The test included a genetic relatedness matrix computed from human genome-wide genotypes (hs GRM), and a genetic relatedness

matrix computed from the parasite genome (*Pf* GRM) either including or excluding the *Pfsa* chromosomes or regions, using the same variants as described for principal components computation (**Supplementary Figure 6**). Columns show: the *Pfsa* region name and variant tested, the population, an indicator of the variants included in the *Pf* GRM, the P-value, estimated effect size, standard error and percentage variance of the parasite genotype explained by HbS; the estimated 'heritability' parameter indicating the contribution of both GRMs to the model fit (on a scale of 0-1); and the estimated mixing parameter determining the relative contribution of the parasite versus the human GRM to the model fit (on a scale of 0-1). We used HbS genotypes called from imputed genotype probabilities with a threshold of 90% certainty, in the 3,346 discovery samples. The analysis is based on 1,960 Gambian and 1,288 Kenyan samples with genotypes meeting this threshold.

1.3) The approach used as evidence for possible co-selection between the *Pfsa* loci used a measure of LD. This is unable to account for population structure, and I would encourage the authors to consider an approach based on mutual information (such as SpydrPick) which can account for relatedness.

We investigated the SpydrPick method but think it is not directly applicable to our data, because it requires as input full genome assemblies or a multiple sequence alignment. (Our data is instead from short read sequencing; producing genome assemblies or an MSA from these data is beyond the scope of our paper.)

In our original manuscript we chose to use the basic Pearson correlation metric r (as well as D' in Supplementary Table 3, now renumbered as Supplementary Table 4) to report LD because it is a simple and widely-used measure of association that many readers are likely to be familiar with. Our results show elevated LD in all sample collections and across all populations where these variants are common, which we think provides convincing evidence this is not a spurious finding. However we agree that understanding the relationship of this LD and population structure is important.

We have therefore now re-estimated LD controlling for population structure (and other covariates) using our software HPTTEST. Specifically, for each pair of *Pfsa* locus lead variants in each population, we fit a logistic regression model with one *Pfsa* variant as a predictor and the second *Pfsa* variant as the outcome variable. To control for population structure we included 20 principal components (PCs) computed from the *P.falciparum* genome as covariates (using the version of PCs computed after excluding the *Pfsa* loci, as described in Supplementary Figure 7). Since this is a logistic regression approach, it generates an estimate of LD expressed as an odds ratio. (In addition to population structure, we also examined other covariates that could in principle affect these estimates, notably including year of admission). This analysis is described in a new Methods section:

“Assessing the influence of covariates on LD estimates

To investigate whether the observed between-locus LD might arise due to population structure effects or due to other artifacts captured by measured covariates in our data, we used HPTTEST to fit a logistic regression model of association with the genotypes at one *Pfsa* locus as outcome and the genotypes at a second *Pfsa* locus as predictor, repeating for each pair of *Pfsa* regions, in each population separately. We fit the model including each of a set of covariates as follows: i. no covariates; ii. 20 parasite principal components; iii. technical covariates including an indicator of the type of sequencing and sequence depth (as in **Supplementary Figure 5**); iv. year of admission, or v. all of the above combined. For each set of covariates we compared the estimated odds ratio indicating the strength of association to the unadjusted odds ratio. In Kenya, across covariate sets, the minimum and unadjusted estimates were 128.0 and 128.0 (*Pfsa*1+ vs *Pfsa*2+; minimum with no

covariates), 218.0 vs. 219.4 (Pfsa1+ vs Pfsa3+; minimum when including technical covariates) and 40.2 vs 47.2 (Pfsa2+ vs Pfsa3+; minimum when including parasite PCs). In Gambia the minimum and unadjusted estimates were 7.0 and 7.7 (Pfsa1+ and Pfsa3+ , minimum when including parasite PCs). These results therefore suggest the observed LD is not substantially explained by population structure or other features of our sample that are captured by these covariates.

We refer to this analysis in main text, where we write:

“This high LD was not explained by population structure or other covariates in our data (Methods), and was observed in multiple populations in MalariaGEN Pf6...”

We hope this addresses this comment.

1.4) More importantly, I didn't think these results were clearly presented by figure 4 (table 1 is good). The main things I think are missing are physical/chromosome distance, and clear identification of the Pfsa loci. I would encourage the authors to investigate a different presentation of these results, rather than the distributions of r. I would suggest a full epistasis/co-selection analysis combined with a circos plot or similar (e.g. fig 2 in Skwark et al 2017 doi:10.1371/journal.pgen.1006508), but I'm sure other presentations are possible.

We have now modified Figure 4 to improve the presentation and to highlight the *Pfsa* alleles. We describe these modifications below before addressing the reviewer's other points. In the revised figure:

- We now plot absolute $|r|$ instead of r (since the sign of r is somewhat arbitrary, depending on the reference genome allele).
- We have now used coloured bars to indicate the LD values due to *Pfsa* variants (and CRT / AAT1 variants in the Gambia), to better demonstrate that the most-outlying LD is substantially due to these regions.
- We have labelled the mutations with highest LD (the *Pfsa* lead variants and the key CRT – AAT1 mutatinos) in both populations.
- The revised Figure also fixes a technical problem with the y axis and bar height in the original figure. The heights now correctly reflect the total number of variant pairs analysed. For clarity we have also stated the variants counts in the legend.

The updated Figure 4 legend now reads::

“Figure 4: HbS-associated variants show extreme between-chromosome correlation in severe *P.falciparum* infections. Histograms show the empirical distribution of absolute genotype correlation ($|r|$, y axis) between pairs of variants on different *Pf* chromosomes in the Gambia (top) and Kenya (bottom). To avoid capturing direct effects of the HbS association, correlation values are computed after excluding HbS-carrying individuals. All pairs of biallelic variants with estimated minor allele frequency at least 5% and at least 75% of samples having non-missing and non-mixed genotype call are shown (totalling 16,487 variants in the Gambia and 13,766 variants in Kenya). Colours indicate the subset of comparisons between HbS-associated variants in *Pfsa* regions relevant for the population (red) and between variants in LD with the CRT K76T mutation (yellow) as shown in the legend. Labelled points denote pairs of regions containing variants with the highest and second highest pairwise correlation in each population; for this purpose regions are defined to include all nearby pairs of correlated variants with minor allele frequency $\geq 5\%$ and $r^2 > 0.05$, such that no other such pair of variants within 10kb of the given region boundaries is present (Methods). A longer list of regions showing elevated between-chromosome LD can be found in **Supplementary Table 5.**”

We think Table 1 is now redundant (as the regions are highlighted in Figure 4, and the details are presented in Supplementary Table 4), so we have removed it from the revised manuscript.

We now address the points relating to within-chromosome distance and Circos-style plots. The aim of Figure 4 is to highlight that the observed LD between the *Pfsa* loci is extremely unusual, and in fact to show that it is qualitatively different from LD at any other loci except the known drug resistance loci. To do this we have taken the simplest approach possible, which is to focus on LD between variants on different chromosomes. This is simplest because – both in theory under neutral models, and as our plot shows also in practice – between-chromosome LD is expected to be close to zero due to independent segregation of chromosomes at meiosis. (By contrast, within-chromosome LD depends on the relative efficiency of recombination between variants and genetic drift (or selection), and any analysis of this would be further complicated by the presence of regions of long LD in the parasite genomes). Because of this focus on between-chromosome pairs, Figure 4 does not depict physical distance between variants.

In preparing our manuscript we also considered using Circos-style plots to show outlying LD pairs, but our preference is not to do this because we think these plots are less informative in practice than our current presentation. Our analysis specifically shows the empirical distribution of LD values for genome-wide variants and highlights the most extreme between-chromosome LD. The distribution of LD is not easily readable from Circos-style plots, and these plots would not make our point in a compelling way.

1.5) The distinction between analysis of multiple populations/countries, and ‘replication’ cohorts (which I believe only include candidate host loci genotyped via sequenom?) were a little unclear. It would be useful to clarify the distinction between replication cohorts and host populations (I found fig S1 helpful, maybe this could be described more in the main text). Or perhaps note the ‘replication’ split by genotyping method is essentially arbitrary, and replication between the Gambia and Kenya is likely more meaningful.

We have reworded our paragraph reporting the discovery and replication analysis to clarify our use of the datasets and refer to the statistical support from both populations, as follows:

“The above results are based on HbS genotypes imputed from surrounding haplotype variation¹, but we focus below on the larger set of 4,071 cases in which we have previously directly assayed HbS genotypes² (**Supplementary Figure 1**). This includes the majority of samples used in our discovery analysis. The *Pfsa1* and *Pfsa3* associations were clearly supported in both populations in this dataset, while *Pfsa2+* appears rare in Gambia (**Supplementary Tables 2 and 3**). We also observed convincing replication of the associations in the additional 825 samples that were not part of our discovery phase, with nominal replication of *Pfsa3* in the Gambia (one-tailed $P=0.026$, $N = 163$) and replication of all three loci in the larger sample from Kenya ($P < 0.001$, $N > 540$) (**Supplementary Table 2**). [...]”

In conjunction with this, we have updated the discovery / replication results in Supplementary Table 2 to include the additional details referred to in the above paragraph. We hope these changes make clear that (from this point on in our manuscript) we are focussing exclusively on the dataset that has directly-typed HbS genotypes, and further clarifies the relationship between the various sets of samples analysed.

We have also updated Supplementary Figure 1 to list the number of replication samples (totaling 825) in the ‘Combined HbS / *Pf* dataset’ box. These match the numbers given in Supplementary Table 2.

1.5b) Figure 2 compounds this problem. It splits the Gambia and Kenya which is useful, but also was at odds with what I understood by 'replication' by this point in the text. A forest plot by cohort would have been better: to this end I found that figure S7 cleared up a lot of my questions and misunderstandings on what was tested and found in each cohort. Perhaps an edited version of fig S7 could be switched with figure 2?

With Figure 2 we are not aiming to establish replication (which has been detailed in the previous paragraphs), but to show our estimate of the protective effect of HbS on specific parasite types. As we set out in Methods these estimates provide additional information on the association (because they are made in reference to population controls and therefore capture relative risks directly). This figure also affords us the opportunity to detail the sample counts. We are keen to do this here because the effect we are reporting is very strong, maybe surprisingly so, and we think the counts are the best way to give transparency on the effect. In particular these counts illustrate our statement that “*the vast majority of children with HbS genotype in our data were infected with parasites that carry Pfsa+ alleles*”.

If it is desirable and space permits we would be happy to explore including a simplified version of Supplementary Figure 7 as a panel in Figure 2. However our feeling is that this has lower importance because, given that the evidence for the association we have presented is convincing (and we think it is, as has also been summarised by the reviewer in comments above), visualising the replication evidence itself is of less interest than detailing the main features of the association, which is what we have aimed to do.

Minor comments:

1.6) In my opinion, Figure 1 and Figure S4 should be switched. Stratifying by host genotype and showing just the pathogen genotype is somewhat subtle to the general reader, and hides the novel interaction approach of this paper.

In the revised manuscript we have replaced Figure 1 with a multi-panel figure which combines the original Figure S4 and Figure 1.

1.7) The authors use a headline figure of 4171 samples in the abstract, and start the results with a number of 5096, but really were only able to analyse 3346 of these with host genotypes. It would be more honest to state 3346 as the number actually used.

We have now updated our abstract as follows:

“[...] In this study we searched for association between candidate host and parasite genetic variants in 3,346 Gambian and Kenyan children ascertained with severe malaria due to *Plasmodium falciparum*. We identified a strong association between sickle haemoglobin (HbS) in the host and three regions of the parasite genome, that is not explained by population structure or other covariates, and that is replicated in additional samples. [...]”

We hope this and our updated wording described in our response to 1.5 above help clarify our use of datasets in the manuscript.

1.8) I think that a simple power analysis for host-pathogen interaction analysis would be very useful for designing future studies of malaria host-pathogen interactions, given the authors are now able to estimate realistic effect sizes.

We have now added a power analysis (Supplementary Figure 12). We describe this analysis and associated additions further in our response to 1.17 below.

1.9) None of the three previous host-pathogen interaction analyses I am aware of have been cited, and probably should be:

Bartha, I. et al. A genome-to-genome analysis of associations between human genetic variation, HIV-1 sequence diversity, and viral control. *Elife* 2, 1–16 (2013)

Azim Ansari, M. et al. Genome-to-genome analysis highlights the effect of the human innate and adaptive immune systems on the hepatitis C virus. *Nat. Genet.* (2017)

doi:10.1038/ng.3835

Lees, J. A. et al. Joint sequencing of human and pathogen genomes reveals the genetics of pneumococcal meningitis. *Nat. Commun.* 10, 2176 (2019) [Noting Col that this is my own paper]

I think from a methodological perspective, contrasting this study with these others would be interesting to statistical genetics readers.

We have now cited these papers in our Methods section:

“We developed a C++ program (HPTEST) to efficiently estimate the odds ratio (4) across multiple human and parasite variants, similar in principle to approaches that have been developed for human-viral and human-bacterial GWAS⁷⁻⁹.”

While we agree a broader methodological comparison between current approaches would be of interest, this is outside the scope of our manuscript which is focussed on the discovery we have made relating to HbS and the *Pfsa* loci.

1.10) Could the authors comment on why *Pfsa2+* is so rarely found in the Gambia, despite apparent co-selection in Kenya?

1.11) Could the low, but constant, frequencies of the alleles be explained by some form of frequency-dependent selection on the parasite?

We think the population genetics of the *Pfsa* variants is extremely intriguing, and that strong selection pressures are evidently operating on these variants. However, because of the range of unusual features present (the three loci involved, the between-locus LD, the observed correlation with HbS frequencies, the fact that *Pfsa2+* is at low frequency in west Africa, and the fact that the allele frequencies don't appear to be changing quickly over the time of our sample) the full nature of this selection is not clear to us – and indeed this will likely be the focus of much future work. In particular some form of frequency dependence between HbS and *Pfsa* variants is clearly present (as shown in Figure 3c) but exactly how this operates is unclear. In the revised manuscript we have further highlighted these issues in our discussion, as follows:

“[...]Given our findings, an obvious hypothesis is that the *Pfsa1+*, *Pfsa2+* and *Pfsa3+* alleles are positively selected in hosts with HbS, but since the frequency of HbS carriers is typically <20%^{2,10} it is not clear whether this alone is a sufficient explanation to account for the high population frequencies or the strong LD observed in non-HbS carriers. Equally, since the *Pfsa+* alleles have not reached fixation (Figure 3) and do not appear to be rapidly increasing in frequency (Supplementary Figure 7), an opposing force may also be operating to maintain their frequency. However, the above data do not suggest strong fitness costs for *Pfsa+*-carrying parasites in HbAA individuals (Figure 2), and the *Pfsa2+* allele also only appears to be present in east Africa, further complicating these observations. [...]”

1.12) The linked resource page <https://www.malariagen.net/resource/32> is empty. MalariaGen are clearly good at sharing information, but I would ask this to be filled out before publication if possible, to assist with review.

We have now updated our Data Availability section to list the available data in more detail:

“Sequence read data from Whole DNA and SWGA sequencing of *P.falciparum* genomes (as detailed in **Supplementary Figure 1**) is available under open-access terms from the European Nucleotide Archive (study accession ERP000190). A full list of relevant sample accessions can be found at <http://www.malariagen.net/resource/32>. Human genotype data used in this study has been described previously^{1,2} and is available under managed-access terms from the European Genome-Phenome Archive (study accession EGAS00001001311), as detailed at <https://www.malariagen.net/resource/25>. A dataset of the human and *Pf* genotypes for 3,346 severe malaria cases used in our discovery scan (**Figure 1**), and HbS genotypes and *Pf* genotypes in the larger set of 4,071 severe cases with direct HbS typing (**Figure 2**) will be made available from Zenodo under open-access terms (doi:10.5281/zenodo.4973476). A full list of data generated by this study and associated resources can be found at <http://www.malariagen.net/resource/32>.”

For full clarity: both the sequence read data and the Zenodo dataset referred to above have been deposited with the respective data repositories. The Zenodo dataset is currently restricted access and will be made fully open prior to publication of this manuscript. (We are happy to give the reviewers or editor access; we think a request via Zenodo is needed for this).

We have also now included a new Supplementary Table (Supplementary Table 8) that lists the relevant *Pf* read data accessions. We have not yet updated the MalariaGEN website but this will be done prior to publication.

1.13) Did the authors consider applying the harmonic mean p-value approach, which can increase power in host-pathogen interaction analysis (Wilson PNAS 2018 doi: 10.1073/pnas.1814092116)? Or is there a reason for not using this here?

We considered the use of the harmonic mean P-value (HMP) in developing Supplementary Figure 4 (now Figure 1a), but we have opted not to use it in our manuscript. The key reasons were:

- The HMP is similar in theory to, and is motivated by, average Bayes factor approaches (while we compute an average Bayes factor directly).
- The HMP did not provide qualitatively different results to our analysis - i.e. that the HbS association is very strong and no other association is compelling in the current analysis.

Our impression was therefore that the HMP is somewhat redundant to our analysis and we chose not to report it.

A more minor reason is that the HMP is theoretically framed around the notion of controlling the family-wise error rate (and more generally around the concept of statistical significance). We think these are not the key quantities of interest – we discuss this further in our response to point 1.17 below.

Typographical/very minor:

1.14) Do the authors state what Pfsa stands for?

In the revised manuscript we have updated our wording to spell out this naming as follows:

“For brevity we shall refer to these HbS-associated loci as *P.falciparum* sickle-associated (Pfsa) 1, 2 and 3 respectively, ...”

1.15) Figure 2 shows a table in a figure, which is difficult to read and extract information from. This is done in a few places, including supplementary figures. Where possible, it would be preferable to add supplementary tables with this information.

In the revised manuscript we have replaced Supplementary Figure 6 with a new supplementary table (Supplementary Table 3). In addition, the counts from Figure 2a are also included in the table.

1.16) The authors talk about two million genotyped *Pf* variants, but analyse just 51225 of them. Some intuition in the main text for this factor of 40 decrease would be appreciated (mostly very rare genotypes?).

Yes, this is due to removing low-frequency variants (as well as multiallelic variants which we found were often less robustly called). This processing is described fully in Supplementary Methods. In the revised manuscript we have clarified this by writing:

“We used an established pipeline to identify and call genotypes at over 2 million single nucleotide polymorphisms (SNPs) and short insertion/deletion variants across the *Pf* genome in these samples (Methods), although the majority of these had low frequency. [...] We focussed on a set of 51,225 biallelic variants in the *P.falciparum* genome that passed all quality control filters and were observed in at least 25 infections in this subset”.

1.17) Noting the multiple testing threshold along with the frequentist results would be useful, given this going to be more stringent than usually expected.

We respond here both to points 1.8 (relating to power analysis), and 1.17.

Appropriate logic for interpreting levels of evidence in genome-wide studies was set out in the original Wellcome Trust Case Control Consortium (WTCCC) GWAS paper¹¹. That analysis makes clear that the evidence for association depends not just on the significance threshold applied to P-values, but also on the association test power and on the prior probability of association, under the relationship:

$$P(\text{association} | p < T) = \frac{\text{prior} \times \text{power}}{\text{prior} \times \text{power} + (1 - \text{prior}) \times T} \quad (2)$$

The WTCCC paper reports this more simply in terms of odds, as:

$$\text{odds}(\text{association} | p < T) = \text{prior odds} \times \frac{\text{power}}{T} \quad (3)$$

The left hand side of formula (2) is the posterior probability of association given observation of a P-value less than the threshold T . (It is equal to one minus the ‘positive false discovery rate’¹²).

The association test power in (2) depends on both the effect size of associated variants and on the variant frequencies. The distribution of true effect sizes is not known, and our

analysis ranges across host and parasite variants which both vary widely in frequency. Consequently there is no choice of T that makes either the power, or the left side of (2), constant – even if we assume a fixed effect size. Power is likely to vary considerably and for this reason we think stating a threshold of this type is counterproductive.

The left-hand-side of (2) also depends on the prior probability of association, which is also unknown (although our study has now shown that it is nonzero.) Assumptions on this prior probability are therefore necessary in interpreting P-values.

(A similar expression to (3) conditional on the observed data also holds; this involves the Bayes factor, whose computation also depends on assuming a distribution of effect sizes. A threshold for the Bayes factor is somewhat simpler to interpret, conditional on this assumed effect size distribution, but interpretation still depends on the prior probability.)

We agree discussion of these points is important, and motivated by these comments we have included both a power analysis, and a discussion of a heuristic approach to determining a prior to use in (2), in our paper. Specifically, our Methods section entitled “Interpretation of association test results” has now been updated to state formula (2) and to describe a heuristic approach to choosing an appropriate prior probability. This reads as follows:

“[...] The 51,552 *Pf* variants represent around 20,000 1kb regions of the *Pf* genome, which might be thought of as approximately independent given LD decay rates¹³; similarly the human genome may be thought of as consisting of around 2 million approximately independent regions. If we take the view that a small number – say up to ten – pairs of regions might be associated, this dictates prior odds on the order of 1 in 4 billion. A Bayes factor around 10^{10} would therefore be needed to generate substantial posterior odds of association, while a Bayes factor an order of magnitude higher would provide compelling evidence (posterior probability > 95%). In **Supplementary Figure 11 and Supplementary Methods** we detail the analogous calculation applied to P-values. For large effect sizes on the order of $OR \sim 4$, this suggests that P-values on the order of $1 \times 10^{-10} - 1 \times 10^{-12}$ might provide compelling evidence for association, depending on the allele frequencies, but weaker effects would require lower thresholds and would be less easily detectable.

It might be considered that the human variants and genes that we have considered here are among those with the highest prior plausibility for association with parasites, and thus the above choice of prior may be considered somewhat conservative. However, even under stronger prior odds on the order of 1 in 2 million (e.g. assuming 10 associations among the variant pairs tested in our study), our results do not identify any associations additional to the HbS-*Pf*sa associations with very strong evidence. Particular variants may however be of further interest due to specific prior plausibility; in **Supplementary Methods** we give further details on putative associations with $BF > 10^5$ and those involving known malaria-protective mutations in the human genome.”

In concert with this we have also added Supplementary Figure 12, which shows an analysis of power and the left hand side of (2) under a range of hypothetical settings. The legend reads:

“**Illustration of association test power and probability of association.** Plot shows approximate association test power (dashed lines) and probability of association (solid lines) for a range of P-value thresholds (x axis) under a range of scenarios (panels and line colours / point shapes). We assume a sample size of 3,346 to match our discovery analysis. The panels vary by prior probability of association (rows) and by the human and parasite variant frequencies (columns), while the line colour and

point shape denotes the assumed association effect size as shown in the legend. The probability of association is computed as $P(\text{association} | p < T)$; the power is defined as $P(p < T | \text{association})$, where T is the given threshold. Results are computed using an approximation to the association test standard error as described in **Supplementary Methods**. P-value thresholds in the range $10^{-10} - 10^{-12}$ provide high probability of association depending on the variant frequencies and effect size, but power is low at these thresholds for rarer variants and for smaller effect sizes.”

1.18) In figure 3, making sure Kenya and the Gambia are labelled in all the panels would be helpful (missing in row 2, panel c).

Thank you. We have updated the figure to include this labelling.

Referee #2 (Remarks to the Author):

This clearly presented manuscript describes an apparent association between Plasmodium falciparum malaria parasite genetic variants and the sickle cell haemoglobin trait (HbS), a trait originally hypothesized to confer resistance to malaria by Tony Allison in 1954 and later experimentally validated by others. As such, this article represents a fascinating update on this story of co-evolution between humans and malaria parasites.

The evidence for this counter-adaptation by malaria parasites comes in multiple forms: 1) Statistical association between the variants in human vs. parasite genomes from sampled infections, 2) an enhanced relative risk of severe malaria in HbS individuals from parasites harboring the candidate variants (albeit in a small sample size), 3) a population-level correlation in the frequency of HbS and the Pfsa1-3+ parasite alleles, and 4) strong linkage disequilibrium (LD), even between chromosomes, for the candidate parasite loci, suggesting co-variation maintained by selection.

While any one of these lines of evidence would be insufficient to warrant confidence in this association, the combined force of all four observations elevates this hypothesis and will motivate followup studies of mechanism and population biology. This work will be of broad interest as an update to a textbook example of host/pathogen co-evolution.

Thank you for these positive comments and the points raised below which we have found helpful.

I have minor questions and clarifications for the authors to address, as follows:

2.1) As the authors note, it is curious that strong LD is maintained between PFsa1/2/3 given modest frequencies of HbS in most populations. Do they exhibit stronger LD in HbS-carriers (AS) individuals than AA individuals? A difference in the magnitude of parasite LD between these host population compartments could be used to infer the magnitude of selection during the establishment of bloodstage infections in HbS individuals.

We have now added the LD estimates and relevant counts, computed across individuals with HbAS/HbSS genotypes, to the revised Supplementary Table 4. In both the Gambia and Kenya, the total count of these individuals is relatively low (19 in the Gambia and 41 in Kenya). In the Gambia, two of the Pfsa+ variants are monomorphic in these individuals, so that the relevant metrics are not defined. However, the estimates can be made in Kenya, where the point estimates of LD are very close to those in HbAA individuals (e.g. $r = 0.82$, 0.76 , and 0.66 for the Pfsa1-2, Pfsa1-3 and Pfsa2-3 comparisons in HbAS / HbSS individuals, versus $r = 0.75$, 0.80 , and 0.66 in HbAA individuals). However, there is a considerable degree of uncertainty in these estimates due to the low sample size.

2.2) Is there an association between Pfsa1/2/3 alleles and HbS in non-severe malaria cases? The apparent abrogation of the protective effect of HbS for severe malaria is interesting, but could the parasite alleles also be enhancing the ability of parasites to establish infections and non-severe malaria cases in HbS carriers?

Addressing this question requires a sample of nonsevere infections in which host genotypes can be determined. To address this, we have now incorporated an analysis of such a sample in the revised manuscript. The data we analyse is from a recent study in which 16 HbAA and 16 HbAS children with uncomplicated malaria were ascertained and parasite RNA-seq data was generated¹⁴. Because the lead Pfsa mutations are in exons, we have been able to use this data to call Pfsa genotypes and to successfully replicate the association analysis in these samples. We refer to these samples in main text, where we write:

“Finally, we analysed available data from a set of uncomplicated infections of Malian children ascertained based on HbS genotype¹⁴ (Methods); this provided further replication of the associations with *Pfsa1* and *Pfsa3* (Supplementary Table 2). Taken together, these data indicate that there are genuine differences in the distribution of parasite genotypes between infections of HbS- and non-HbS genotype individuals.”

The details of this analysis are provided in the revised Supplementary Table 2 (association test results) and in the newly added Supplementary Table 6 (which gives full details of the samples and genotypes).

Although the sample size is modest, this newly added data does suggest that the HbS-*Pfsa* association carries over to nonsevere infections as well.

2.3) Are there any hypotheses to explain why the HbS/MSP1 association was not replicated in this study? This deserves some comment.

We have added a new Supplementary Text section to specifically address this. There are several possibilities. The original finding arose from a relative small study (N=77 and N=163 in the two cohorts analysed) and had a number of complicating factors; thus false positives are a possibility given lack of replication in our larger study. However, the MSP1 locus harbours complex genetic variation and this is likely not fully accessed using our approach based on biallelic SNPs, so it is also possible that we do not fully test this in our study.

*2.4) In Figure 1 it is evident that multiple variants are associated with HbS at each parasite locus. Do *Pfsa1/2/3* show evidence of recent positive selection in the parasite genome (ie long haplotypes from sweeps), or do they look like old, balanced polymorphisms? Does *Pfsa2*, being more limited in geographic distribution, sit on a longer haplotype?*

We have now revised our discussion to clarify our thoughts on selection. We write (changes highlighted in yellow):

“Taking together these new findings with other population genetic evidence from multiple locations across Africa, including observations of frequency differentiation within and across *P.falciparum* populations^{13,15,16} and other metrics at these loci indicative of selection¹⁷⁻¹⁹, it appears likely that the allele frequencies and strong linkage disequilibrium between *Pfsa1*, *Pfsa2* and *Pfsa3* are maintained by a form of natural selection. However, the mechanism for this is unclear. Given our findings, an obvious hypothesis is that the *Pfsa1+*, *Pfsa2+* and *Pfsa3+* alleles are positively selected in hosts with HbS, but since the frequency of HbS carriers is typically <20%^{2,10} it is not clear whether this alone is a sufficient explanation to account for the high population frequencies or the strong LD observed in non-HbS carriers. Equally, since the *Pfsa+* alleles have not reached fixation (Figure 3) and do not appear to be rapidly increasing in frequency (Supplementary Figure 7), it may be presumed that an opposing force is operating; but the above data do not suggest strong fitness costs for *Pfsa+*-carrying parasites in HbAA individuals (Figure 2). The *Pfsa2+* allele also only appears to be present in east Africa, further complicating these observations. It thus remains entirely possible that additional selective factors are involved, such as epistatic interactions between these loci, or further effects on fitness in the host or vector in addition to those observed here in relation to HbS.”

Specifically relating to recent positive selection, the *Pfsa* variants have not arisen in scans for selective sweeps in most African populations¹⁷ (although some evidence was observed for a sweep near PfACS8 in a small sample from Senegal¹⁸). The variants also do not

appear to change rapidly in frequency over time in our sample (Supplementary Figure 7), and clearly have frequencies that relate to HbS frequencies (Figure 3c). The *Pfsa3+* variant also appears on a structural variant haplotype that is shared by multiple populations (Supplementary Figure 11). We therefore believe the *Pfsa* loci have not generally undergone strong recent selective sweeps in most populations. As the paragraph above further sets out, there are a number of other intriguing features of these loci which indicate they are under a form of selection, but we do not currently understand what the relevant forces (other than HbS) are, or how they might be interacting to determine present-day frequencies.

Referee #3 (Remarks to the Author):

3.1 This is a strong paper with many interesting findings that obviously have high scientific and indeed clinical importance. The authors find good evidence that there are polymorphisms in the malaria genome that lead to more severe disease in individuals with the sickle cell genotype. These polymorphisms are more common in regions where the sickle cell genotype is more common and are in unexpectedly strong linkage disequilibrium with each other, given they are on different chromosomes.

3.2. The complicating factor in this analysis is population structure. It is quite hard to get one's head around all the possibilities, since there is human population structure and parasite population structure, which can be correlated with each other, via geography or indeed via natural selection and there can also be differences in the rate of severe or moderate malarial disease that are due to human factors that are associated with geography. One of the disease loci does contribute to an important PC, which does reduce the association signal between the locus and severe malaria, but presumably does leave the overall set of associations intact.

3.4. I would say that population structure makes interpretation of any one part of the results more difficult but that the patterns are very striking. I think the paper as it stands gets the balance right in terms of highlighting the caveats, while making the observations and their prima facie interpretations clear. The main text figures represent a good presentation of the basic underlying data. More elaborate figures are possible but I am not sure they would be beneficial.

Thank you for your review. These comments align with our understanding, which is that while subtle population structure effects might be possible in general, they are clearly not the cause of the HbS-*Pfsa* association we have reported, which is extremely strong. In our presentation we have aimed to keep the figures as simple as possible to describe the key features of this association.

3.5 So in sum, I do not have any major suggestions. An extremely minor one is that in Figure 2, (controls) is not properly aligned, which did actually make it a little hard for me to understand the figure when I first looked at it.

We have updated Figure 2 to correct this misalignment. (Please see our response to 4.4 below for additional changes we have made to this figure.)

Referee #4 (Remarks to the Author):

In this manuscript, the authors estimate associations between human loci that have been associated with risk of severe malaria and parasite genotypes, assuming an additive interaction. They identify three parasite loci, in linkage disequilibrium with each other, that are significantly associated with presence of HbS, including two genes on chromosome 2 and one on chromosome 11. While almost all infections in HbS individuals contained mutations at these loci, these mutations were also common in non-HbS individuals, including individuals without severe malaria. The study generates interesting hypotheses that could be explored further, even in the context of this study, but that will require functional validation to determine mechanisms and the impact of the findings. Specific comments are listed below.

4.1 The authors provide only one paragraph describing the premise behind the study, what is already known about the research question, and what the knowledge gaps are. This short paragraph did not convey the potential impact of the study and why they are doing it.

In our introduction we briefly outline that human genetic variants including HbS are known to convey protection, and that parasites are known to be genetically diverse and to be able to evolve rapidly in response to selection pressure. This directly raises the question of whether parasites might evolve to overcome the natural protection, and this is the central premise of our paper. We think this general theme of a human-malaria 'arms race' will be recognised by most readers, and because of this we have deliberately kept this paragraph short.

To provide extra clarity on this, we have updated our paragraph describing our approach to emphasise that we are examining variants that plausibly are involved in resistance in this way:

*“We used a logistic regression approach to test for pairwise association between these *P. falciparum* variants and four categories of human variants that are plausibly associated with malaria resistance:[...]”*

Given the other requested additions we would prefer to preserve the brevity of the initial paragraph if the reviewer and editor are happy with that approach.

4.2 It would be helpful for the authors to include what is now Supplementary Figure 1 in the main text, as it is difficult to follow the source of different samples, which underwent direct WGS versus sWGA followed by WGS, which had genome-wide human data versus genotyping, etc. In addition, including a sentence or two briefly describing the criteria for removing closely related individuals would be helpful.

To clarify our use of the different datasets in the paper, we have now updated our wording to make clear that (following our initial discovery analysis) we focus throughout on the dataset with directly-typed HbS genotypes:

“The above results are based on HbS genotypes imputed from surrounding haplotype variation¹, but we focus below on the larger set of 4,071 cases in which we have previously directly assayed HbS genotypes² (Supplementary Figure 1). This includes the majority of samples used in our discovery analysis. [...]”

Supplementary Figure 1 is a large flowchart diagram that provides full detail on how the samples from our data were processed. We appreciate the above point but have decided to keep this figure in supplementary, since it is a detailed technical figure relating to data generation. Although the data provenance is important, we don't think these technical aspects play a large role in our results in practice, in the sense that (as we demonstrate) the effect we describe is not confounded by technical aspects of the sequencing. It is also a sufficiently strong effect that it would be detectable in any reasonably large dataset of this type. We hope that Supplementary Figure 1 and the other supplementary material is well cited in the main text, such that it can easily be located by readers who are interested in the details of the data generation and methods.

4.3 Can the authors explain why they looked at an additive interaction between loci? Often a multiplicative model is used to assess interaction terms for measures of effect like OR and RR.

We have updated the paper to clarify our use of the word ‘additive’ as follows:

“...focussing on the variant with the strongest association in each region and assuming an additive model of effect of the host allele on parasite genotype on the log-odds scale”

The word ‘additive’ here refers to the way in which the three human genotypes (homozygote and heterozygote) affect the probability of carrying the outcome (i.e. the *Pf* alternative allele). Because this is logistic regression, this is modelled as additive on the log-odds scale, i.e. the log-odds conferred by the heterozygote genotype is half way between that of the two homozygote genotypes. On the odds scale, the model is multiplicative, so we think this does align with the reviewer’s expectation.

4.4 Likewise, for the analysis of the effect of HbS on severe malaria presented in Figure 2b, it might be better to construct a model including an interaction term representing the Pfsa+ loci, rather than presenting a stratified analysis with very small numbers of individuals for comparison. Also, was parasitemia included as a potential confounding variable, or was this variable used in the definition of severe malaria?

We agree that some strata in this Figure 2b are based on low counts which makes the corresponding estimates have considerable uncertainty. This is visually represented in the wide intervals for these estimates, but the reviewer’s comment has highlighted to us that these estimates are overemphasised in the figure. In the revised manuscript we have therefore updated this figure as follows:

1. The point size now varies with the number of informative samples (i.e. the number of samples with HbAS or HbAA genotype and the given parasite genotype), which we think improves the figure by placing the emphasis on the more accurate estimates.
2. We have also taken the opportunity to make a slight technical correction to the credible intervals shown in Figure 2b: they are now based on the 2.5% and 97.5% estimated posterior quantiles (as opposed to the posterior mean and standard deviation as previously). This gives similar results but means they better reflect the posterior, and are no longer exactly centred on the point estimates.

The legend has been updated accordingly to read:

“Relative risks were estimated using a multinomial logistic regression model [..]. Circles reflect posterior mean relative risk estimates and horizontal lines reflect the corresponding 95% credible intervals. Estimates based on < 5 individuals with HbAS or HbSS genotypes are represented by smaller circles. [...]”

We thank the reviewer for this suggestion as we think this improves Figure 2b. We now turn to the other points raised. We think the reviewer’s suggestion is to fit a model in which the human and parasite genotypes, and an interaction term, are treated as predictors, i.e. of the form:

$$\text{severe malaria status} \sim \text{human genotype} + \text{Pf genotype} + \text{interaction terms} \quad (2)$$

Unfortunately this model cannot be fit in our data. The reason is that the controls used in Figure 2 are population controls – i.e. they are a sample from the general population. They do not have measured parasite genotypes which would be necessary to fit a model of the form (2).

However, the aim of Figure 2b is different from the model (2): it aims to show the estimated level of protection afforded by HbS against severe malaria. We think this is the right way to present the data for two reasons. First, the figure is on the scale of population relative risk, and this makes the estimates easily interpretable. For example they are directly comparable to previous estimates of the protective effect of HbS on severe malaria (often estimated at around RR ~ 0.1 e.g. as in ²⁰). Second, as we describe in Methods these relative risks are also closely related to the association odds ratios as estimated in severe malaria cases, forming a link to the association analysis described further up the paper. Consequently we think presenting these relative risks is a very natural way to describe the data that fits well into our overall narrative.

We agree it would be interesting to fit model (2); this would require using a sample of non-severe (or asymptomatic) infections to use as controls and to avoid confounding these would need to be collected by the same a similar sampling scheme to the severe cases. We are currently not aware of a sample of severe and nonsevere cases in which this model could be fit at present. However, we note that (if such a sample could be generated), fitting (2) would also answer a different question to that presented in Figure 2, since the results would relate to the difference between nonsevere and severe infections.

All of the severe malaria samples analysed in our study were 1. collected with symptoms of severe malaria and 2. had sufficient parasitaemia that high-quality genotype calls were possible from genome sequencing. Thus, parasitaemia (presence of parasites) was a precondition for inclusion in our study.

4.5 Although I like that Figure 1 shows the context of where the associated loci are with respect to other genes in the genome, I find supplementary Figure 4 to be more comprehensive and visually appealing.

In the revised manuscript we have now combined Supplementary Figure 4 and Figure 1 in a single multi-panel figure.

4.6 Did the authors estimate the association between the Pfsa+ loci and severe malaria in individuals without HbS? Also, the frequency of Pfsa+ alleles seems similar between the severe cases and the population controls from MalariaGEN Pf6K (as noted by the authors). The data from Pf6K presumably primarily represent cases of uncomplicated malaria. Is it possible that these loci are simply virulence factors that increase the likelihood of clinical malaria, regardless of HbS status?

We did not directly estimate the association between parasite genotypes and severe malaria. The reason is that (as with the interaction model above) this would require a set of nonsevere infections, e.g. uncomplicated or asymptomatic cases) to use as controls. Our dataset does not contain such controls and we are not aware of a dataset where this comparison could be carried out currently (although we agree it would be worthwhile).

As the reviewer points out, a rough comparison can be made between Pf6 and our severe case data. This does not support a full statistical analysis because the sampling of these samples was different from that used for the severe cases in our studies (including different collection years). However, if we ignore this concern, Figure 3 shows that the Pfsa+ allele frequencies appear at slightly lower frequency in severe cases than in the Pf6 (uncomplicated) cases in both Kenya and The Gambia. Our interpretation is that these alleles therefore do not strongly associate with virulence (i.e. do not strongly tend to cause severe disease rather than less severe symptoms). However as described above we think further samples will be needed to fully resolve this.

4.7 Figure 3c represents an ecological comparison that does not represent associations at an individual level; therefore, it should be interpreted cautiously and not be used to attribute a causal association.

We use Figure 3 to illustrate a descriptive analysis of the population frequencies of the alleles. We write:

“The *Pfsa1+*, *Pfsa2+* and *Pfsa3+* alleles had similar frequencies in Kenya (approximately 10-20%) whereas in Gambia *Pfsa2+* had a much lower allele frequency than *Pfsa1+* or *Pfsa3+* (< 3% in all years studied, versus 25-60% for the *Pfsa1+* or *Pfsa3+* alleles; **Figure 3a**). To explore the population genetic features of these loci in more detail, we analysed the MalariaGEN Pf6 open resource which gives *P. falciparum* genome variation data for 7,000 worldwide samples ²¹ (**Figure 3b**). This showed considerable variation in the frequency of these alleles across Africa, the maximum observed value being 61% for *Pfsa3+* in the Democratic Republic of Congo, and indicated that these alleles are rare outside Africa. Moreover, we found that within Africa, population frequencies of the *Pfsa+* alleles are strongly correlated with the frequency of HbS (**Figure 3c**, estimated using data from the Malaria Atlas Project ¹⁰).”

This paragraph does not draw causal conclusions.

We also refer to this figure in our discussion, where we highlight the features of these alleles that suggest they are evolving under natural selection - in particular that the allele frequencies are correlated with, but at higher frequency than, that of HbS. In the revised manuscript we have further updated this discussion as follows:

“Given our findings, an obvious hypothesis is that the *Pfsa1+*, *Pfsa2+* and *Pfsa3+* alleles are positively selected in hosts with HbS, but since the frequency of HbS carriers is typically <20% ^{2,10} it is not clear whether this alone is a sufficient explanation to account for the high population frequencies or the strong LD observed in non-HbS carriers. Equally, since the *Pfsa+* alleles have not reached fixation (**Figure 3**) and do not appear to be rapidly increasing in frequency (**Supplementary Figure 7**), it may be presumed that an opposing force is operating; but the above data do not suggest strong fitness costs for *Pfsa+*-carrying parasites in HbAA individuals (**Figure 2**). The *Pfsa2+* allele also only appears to be present in east Africa, further complicating these observations.”

As the paragraph indicates, we think Figure 3c (along with the other features of these loci) clearly point at strong selection effects on these loci, but the set of forces involved and the full mechanism are currently unclear to us.

4.8 It would be helpful for the authors to delve more into the biological plausibility of the three loci identified in this study in modifying the effect of HbS on severe malaria. There is very little discussion of the proteins encoded by these genes. What are the predicted functions (GO terms), essentiality, pathways, timing of expression, etc.? How might this information support the contribution of these loci to severe malaria? Much of this information can be found in plasmDB. For example, the gene Pf3D7_0220300 has been identified as a candidate gene for virulence and encodes an exported protein. The functional studies may be beyond the scope of this paper (although such experiments would greatly strengthen the paper for publication in a high impact journal such as Nature), but currently there is very little interpretation of the results in the manuscript as written.

We agree, and to address this comment we have added a substantial new analysis of available functional data to the revised manuscript. This analysis is in two parts. We first

examined what is known about expression of the relevant genes, and the function and localisation of the corresponding proteins. We summarise this in an updated paragraph as follows:

“The biological function of these parasite loci is a matter of considerable interest for future investigation. At the *Pfsa1* locus, the signal of association includes non-synonymous changes in the *PfACS8* gene, which encodes an acyl-CoA-synthetase²². It belongs to a gene family that has expanded in the Laverania relative to other *Plasmodium* species²³, and lies close to a paralog *PfACS9* on chromosome 2. *PfACS8* has been predicted to localise to the apicoplast²⁴, but it also contains a PEXEL motif^{25,26} which may instead indicate export to the host cytosol (where other ACS family members have been observed²⁷). The functions of the proteins encoded by *PF3D7_0220300* (an exported protein, at the *Pfsa2* locus) and *PF3D7_1127000* (a putative tyrosine phosphatase, at *Pfsa3*) are not known; however, *PF3D7_0220300* has been observed to localise to the host membrane and to colocalise with host stomatin²⁸, while *PF3D7_1127000* has been observed in the food vacuole²⁹. All three genes appear to be expressed at multiple parasite lifecycle stages (**Supplementary Text**) in 3D7 parasites, in particular at ring stage (for *PfACS8*) and trophozoite stage (for *PF3D7_0220300* and *PF3D7_1127000*). They have not been found essential for *in vitro* growth³⁰.”

The details form a new Supplementary Text section (Supplementary Text 2.2.1-2.2.4).

We have then gone on to analyse possible functional implication of the *Pfsa* variants themselves. This analysis revealed two features which we think may be significant. First, we noted that two of the *Pfsa+* alleles lie immediately downstream of PEXEL motifs (i.e. of key amino acid sequences that are known to mediate protein export from the parasite to the erythrocyte cytosol). This suggests these variants may be affecting protein export processes. But secondly, we also show that there is an apparent effect of the *Pfsa3+* allele on gene expression. Specifically *Pfsa3+* is associated with increased expression of *Pf3D7_1127000* at trophozoite stage. This is supported by data from a recent study from Mali¹⁴, which we have now reanalysed in our paper. These results are shown in new Supplementary Figures 9 and 10) and detailed in a new paragraph in main text, which reads:

“We noted two further features that may point to the functional role of the *Pfsa+* alleles themselves. The associated variants at *Pfsa2* and *Pfsa3* each include SNPs immediately downstream of a PEXEL motif (detailed in **Supplementary Text**), which mediates export through a pathway that involves protein cleavage at the motif³¹. This process leaves the downstream amino acids at the N-terminal of the mature protein, and their sequence has been found to influence successful export^{32,33}. It is therefore possible that these alleles affect export of the corresponding proteins. However, an alternative possibility is that the *Pfsa+* alleles affect levels of transcription of the relevant genes. In this context, we noted a recent study which found that *PF3D7_1127000* is among the most differentially over-expressed genes in trophozoite-stage infections of HbAS compared to HbAA children (>32-fold increase in transcripts per million (TPM) at trophozoite stage; N = 12; unadjusted P = 5.6x10⁻²²; using¹⁴). We reanalysed this data in light of genotypes at the *Pfsa* loci (**Supplementary Table 6**), and found that the *Pfsa3+* mutations plausibly explain this increased expression. Specifically, read ratios at the second-most associated *Pfsa3* SNP (chr11:1,057,437 T > C) (**Supplementary Table 1**) appear particularly strongly correlated with increased expression at trophozoite stage (**Supplementary Figure 8**; underlying data shown in **Supplementary Table 6**). Further support for this observation comes from an *in vitro* time course experiment conducted in the same study¹⁴ in which the increased expression is observed in AA erythrocytes infected with a *Pfsa+*-carrying isolate (**Supplementary Figure 9-10** and **Methods**). The mechanism of upregulation is not known, but a further relevant observation is that the *Pfsa3+* alleles appear linked to a neighbouring copy number variant that includes

duplication of the 5' end of the small nuclear ribonucleoprotein *SNRPF* upstream of 1127000 (based on analysis of available genome assemblies of *P. falciparum* isolates³⁴; **Supplementary Figure 10**). We caution that these findings are tentative, and the manner in which *Pfsa* alleles affect genome function is a matter for future research.

Understanding this functional role could provide important clues into how HbS protects against malaria and help to distinguish between the various proposed mechanisms, which include enhanced macrophage clearance of infected erythrocytes³⁵, inhibition of intraerythrocytic growth dependent on oxygen levels³⁶, altered cytoadherence of infected erythrocytes³⁷ due to cytoskeleton remodelling³⁸ and immune-mediated mechanisms³⁹.”

In support of the above we have added three new Supplementary Figures (Supplementary Figures 8-10) and new Supplementary Text sections 2.2.5-2.2.6.

As the paragraph above emphasises, while these are important leads, we nevertheless think that fully uncovering the biological function of these loci is likely to take considerable future effort. Indeed part of the interest of our finding will be to prioritise these relatively unknown regions for future investigation. We hope the current changes address the reviewer's comment. Thank you for the very helpful review.

References

- 1 Band, G. *et al.* Insights into malaria susceptibility using genome-wide data on 17,000 individuals from Africa, Asia and Oceania. *Nature Communications* **10**, 5732, doi:10.1038/s41467-019-13480-z (2019).
- 2 Malaria Genomic Epidemiology Network. Reappraisal of known malaria resistance loci in a large multicenter study. *Nat Genet* **46**, 1197-1204, doi:10.1038/ng.3107 (2014).
- 3 Timmann, C. *et al.* Genome-wide association study indicates two novel resistance loci for severe malaria. *Nature* **489**, 443-446, doi:10.1038/nature11334 (2012).
- 4 Leffler, E. M. *et al.* Resistance to malaria through structural variation of red blood cell invasion receptors. *Science* **356**, doi:10.1126/science.aam6393 (2017).
- 5 Cowman, A. F., Berry, D. & Baum, J. The cellular and molecular basis for malaria parasite invasion of the human red blood cell. *J Cell Biol* **198**, 961-971, doi:10.1083/jcb.201206112 (2012).
- 6 Cowman, A. F., Tonkin, C. J., Tham, W. H. & Duraisingh, M. T. The Molecular Basis of Erythrocyte Invasion by Malaria Parasites. *Cell Host Microbe* **22**, 232-245, doi:10.1016/j.chom.2017.07.003 (2017).
- 7 Ansari, M. A. *et al.* Genome-to-genome analysis highlights the effect of the human innate and adaptive immune systems on the hepatitis C virus. *Nature Genetics* **49**, 666-673, doi:10.1038/ng.3835 (2017).
- 8 Bartha, I. *et al.* A genome-to-genome analysis of associations between human genetic variation, HIV-1 sequence diversity, and viral control. *Elife* **2**, e01123, doi:10.7554/eLife.01123 (2013).
- 9 Lees, J. A. *et al.* Joint sequencing of human and pathogen genomes reveals the genetics of pneumococcal meningitis. *Nature Communications* **10**, 2176, doi:10.1038/s41467-019-09976-3 (2019).
- 10 Piel, F. B. *et al.* Global epidemiology of sickle haemoglobin in neonates: a contemporary geostatistical model-based map and population estimates. *Lancet* **381**, 142-151, doi:10.1016/S0140-6736(12)61229-X (2013).
- 11 Wellcome Trust Case Control Consortium. Genome-wide association study of 14,000 cases of seven common diseases and 3,000 shared controls. *Nature* **447**, 661-678, doi:10.1038/nature05911 (2007).
- 12 Storey, J. D. The positive false discovery rate: a Bayesian interpretation and the q -value. *The Annals of Statistics* **31**, 2013-2035, doi:10.1214/aos/1074290335 (2003).
- 13 Pearson, R. D., Amato, R. & Kwiatkowski, D. P. An open dataset of Plasmodium falciparum genome variation in 7,000 worldwide samples. *bioRxiv*, 824730, doi:10.1101/824730 (2019).
- 14 Saelens, J. W. *et al.* Impact of sickle cell trait hemoglobin on the intraerythrocytic transcriptional program of Plasmodium falciparum. *bioRxiv*, 2021.2008.2006.455439, doi:10.1101/2021.08.06.455439 (2021).
- 15 Moser, K. A. *et al.* Describing the current status of Plasmodium falciparum population structure and drug resistance within mainland Tanzania using molecular inversion probes. *Molecular Ecology* **30**, 100-113, doi:<https://doi.org/10.1111/mec.15706> (2021).
- 16 Verity, R. *et al.* The impact of antimalarial resistance on the genetic structure of Plasmodium falciparum in the DRC. *Nature Communications* **11**, 2107, doi:10.1038/s41467-020-15779-8 (2020).
- 17 Amambua-Ngwa, A. *et al.* Major subpopulations of Plasmodium falciparum in sub-Saharan Africa. *Science* **365**, 813-816, doi:10.1126/science.aav5427 (2019).
- 18 Chang, H.-H. *et al.* Genomic Sequencing of Plasmodium falciparum Malaria Parasites from Senegal Reveals the Demographic History of the Population. *Molecular Biology and Evolution* **29**, 3427-3439, doi:10.1093/molbev/mss161 (2012).

- 19 Park, D. J. *et al.* Sequence-based association and selection scans identify drug resistance loci in the *Plasmodium falciparum* malaria parasite. *Proceedings of the National Academy of Sciences* **109**, 13052, doi:10.1073/pnas.1210585109 (2012).
- 20 Taylor, S. M., Parobek, C. M. & Fairhurst, R. M. Haemoglobinopathies and the clinical epidemiology of malaria: a systematic review and meta-analysis. *Lancet Infect Dis* **12**, 457-468, doi:10.1016/S1473-3099(12)70055-5 (2012).
- 21 Ahouidi, A. *et al.* An open dataset of *Plasmodium falciparum* genome variation in 7,000 worldwide samples [version 1; peer review: awaiting peer review]. *Wellcome Open Research* **6**, doi:10.12688/wellcomeopenres.16168.1 (2021).
- 22 Matesanz, F., Téllez, M. a.-d.-M. & Alcina, A. The *Plasmodium falciparum* fatty acyl-CoA synthetase family (PfACS) and differential stage-specific expression in infected erythrocytes. *Molecular and Biochemical Parasitology* **126**, 109-112, doi:[https://doi.org/10.1016/S0166-6851\(02\)00242-6](https://doi.org/10.1016/S0166-6851(02)00242-6) (2003).
- 23 Otto, T. D. *et al.* Genomes of all known members of a *Plasmodium* subgenus reveal paths to virulent human malaria. *Nature Microbiology* **3**, 687-697, doi:10.1038/s41564-018-0162-2 (2018).
- 24 Ralph, S. A. *et al.* Metabolic maps and functions of the *Plasmodium falciparum* apicoplast. *Nature Reviews Microbiology* **2**, 203-216, doi:10.1038/nrmicro843 (2004).
- 25 Hiller, N. L. *et al.* A host-targeting signal in virulence proteins reveals a secretome in malarial infection. *Science* **306**, 1934-1937, doi:10.1126/science.1102737 (2004).
- 26 Marti, M., Good, R. T., Rug, M., Knuepfer, E. & Cowman, A. F. Targeting malaria virulence and remodeling proteins to the host erythrocyte. *Science* **306**, 1930-1933, doi:10.1126/science.1102452 (2004).
- 27 Matesanz, F., Durán-Chica, I. & Alcina, A. The cloning and expression of Pfacs1, a *Plasmodium falciparum* fatty acyl coenzyme A synthetase-1 targeted to the host erythrocyte cytoplasm¹¹ Edited by M. Yaniv. *Journal of Molecular Biology* **291**, 59-70, doi:<https://doi.org/10.1006/jmbi.1999.2964> (1999).
- 28 Butler, T. K. An Exported Malaria Protein Regulates Glucose Uptake During Intraerythrocytic Infection. *Washington University in St. Louis PhD Thesis* (2014).
- 29 Lamarque, M. *et al.* Food vacuole proteome of the malarial parasite *Plasmodium falciparum*. *PROTEOMICS – Clinical Applications* **2**, 1361-1374, doi:<https://doi.org/10.1002/prca.200700112> (2008).
- 30 Zhang, M. *et al.* Uncovering the essential genes of the human malaria parasite *Plasmodium falciparum* by saturation mutagenesis. *Science* **360**, eaap7847, doi:10.1126/science.aap7847 (2018).
- 31 Russo, I. *et al.* Plasmepsin V licenses *Plasmodium* proteins for export into the host erythrocyte. *Nature* **463**, 632-636, doi:10.1038/nature08726 (2010).
- 32 Boddey, J. A. *et al.* Role of Plasmepsin V in Export of Diverse Protein Families from the *Plasmodium falciparum* Exportome. *Traffic* **14**, 532-550, doi:<https://doi.org/10.1111/tra.12053> (2013).
- 33 Grüring, C. *et al.* Uncovering Common Principles in Protein Export of Malaria Parasites. *Cell Host & Microbe* **12**, 717-729, doi:<https://doi.org/10.1016/j.chom.2012.09.010> (2012).
- 34 Otto, T. D. *et al.* Long read assemblies of geographically dispersed *Plasmodium falciparum* isolates reveal highly structured subtelomeres. *Wellcome Open Res* **3**, 52, doi:10.12688/wellcomeopenres.14571.1 (2018).
- 35 Luzzatto, L. Sick cell anaemia and malaria. *Mediterr J Hematol Infect Dis* **4**, e2012065, doi:10.4084/MJHID.2012.065 (2012).
- 36 Archer, N. M. *et al.* Resistance to *Plasmodium falciparum* in sickle cell trait erythrocytes is driven by oxygen-dependent growth inhibition. *Proceedings of the National Academy of Sciences* **115**, 7350-7355, doi:10.1073/pnas.1804388115 (2018).

- 37 Cholera, R. *et al.* Impaired cytoadherence of *Plasmodium falciparum*-infected erythrocytes containing sickle hemoglobin. *Proc Natl Acad Sci U S A* **105**, 991-996, doi:10.1073/pnas.0711401105 (2008).
- 38 Cyrklaff, M. *et al.* Hemoglobins S and C Interfere with Actin Remodeling in *Plasmodium falciparum*-Infected Erythrocytes. *Science* **334**, 1283-1286, doi:10.1126/science.1213775 (2011).
- 39 Williams, T. N. *et al.* An immune basis for malaria protection by the sickle cell trait. *PLoS Med* **2**, e128, doi:10.1371/journal.pmed.0020128 (2005).

Reviewer Reports on the First Revision:

Referee #1 (Remarks to the Author):

Overall this is a thorough response to my comments, clearly showing the effort and attention to detail the authors have spent on this paper. Also I thought that the addition of functional interpretation in response to one of the other reviewer's comments was particularly helpful. I have no further substantive comments, but have a few 'responses to responses' that the authors may consider when deciding what to include in the final version.

Using the numbering in the authors' responses:

1.1

It is fair enough that there is already plenty in this paper, and a full genome-to-genome analysis could certainly be standalone work. I would really like more of a clarification to the text to this issue than has been added though - adding a sentence or two more on motivation would help readers follow the study design a lot more easily [see also response 4.1]. Particularly, from the authors' response:

'We deliberately aim to test a strong biological hypothesis, specifically that malaria parasites may have evolved to overcome resistance conferred by naturally occurring protective host mutations. This motivates looking for association between *Plasmodium falciparum* (Pf) variation and the host loci that might be plausibly involved in these interactions.'

Could be added to line 78 to clarify why other markers were ignored (for now). You may also consider adding a sentence to the discussion that a full genome-to-genome analysis would be possible in future, but was beyond the scope of this study.

1.2

I found this response particularly interesting, and thank the authors for such a detailed reply. Overall, I am happy with the result that this analysis did not change the biological finding in the paper. Personally I found this discussion very useful from a technical standpoint, but do appreciate that using just these p-values as the main results would add confusion. I would be in favour of adding this description and table R1 somewhere to the supplementary materials for interested readers (as such host/pathogen studies are likely to become more common), but leave this decision to the authors.

I also note that there are implementations of linear mixed models which have Bernoulli distributed errors (with logit or probit link) which could be used to further test the theory that differences are down to model misspecification, but I certainly am not asking the authors to do this analysis here, as they have answered my question thoroughly already.

1.4

Thank you for looking into a circos plot / epistasis analysis, and sorry that it was a dead end. Figure 4 is easier to understand now.

John Lees

Referee #2 (Remarks to the Author):

I am completely satisfied with the thorough response of the authors to my comments and those of the other reviewers.

Referee #4 (Remarks to the Author):

Most of the comments from the initial review have been adequately addressed; however, the authors should consider making it explicit for the reader that figure 3 does not show individual-level comparisons. In saying that the comparison is "ecological", it was meant that the authors should make it clear that although the populations with high HbS frequency also tend to have high frequency of the parasite alleles of interest, this comparison does not indicate that the HbS individuals were more likely to harbor parasites with the alleles of interest.

Author Rebuttals to First Revision:

We would like to thank all of the reviewers for their comments which have led to substantial improvements to our manuscript.

Referee #1 (Remarks to the Author):

Overall this is a thorough response to my comments, clearly showing the effort and attention to detail the authors have spent on this paper. Also I thought that the addition of functional interpretation in response to one of the other reviewer's comments was particularly helpful. I have no further substantive comments, but have a few 'responses to responses' that the authors may consider when deciding what to include in the final version.

Using the numbering in the authors' responses:

1.1

It is fair enough that there is already plenty in this paper, and a full genome-to-genome analysis could certainly be standalone work. I would really like more of a clarification to the text to this issue than has been added though - adding a sentence or two more on motivation would help readers follow the study design a lot more easily [see also response 4.1]. Particularly, from the authors' response:

'We deliberately aim to test a strong biological hypothesis, specifically that malaria parasites may have evolved to overcome resistance conferred by naturally occurring protective host mutations. This motivates looking for association between Plasmodium falciparum (Pf) variation and the host loci that might be plausibly involved in these interactions.'

Could be added to line 78 to clarify why other markers were ignored (for now). You may also consider adding a sentence to the discussion that a full genome-to-genome analysis would be possible in future, but was beyond the scope of this study.

We have reviewed our text around this in light of this comment. In the previous manuscript revision:

- We refer to our choice to focus on a set of candidate loci in the second sentence of the abstract, writing "In this study we searched for association between candidate host and parasite genetic variants in 3,346 Gambian and Kenyan children ascertained with severe malaria due to *Plasmodium falciparum*"
- The first paragraph of the manuscript sets out our central hypothesis, ending with what we refer to as the "basic question": "...are there genetic forms of *P. falciparum* that can overcome the human variants that confer resistance to this parasite?"

- The third paragraph sets out clearly that we are focussing on specific categories of human mutations motivated by the above (“*We used a logistic regression approach to test [with] four categories of human variants that are plausibly associated with malaria resistance: ...*”) and goes on to specify what these sets are.

This wording was the result of changes made in light of your earlier comments and those of other reviewers, and we do think it is substantially clearer than the originally submitted version. On balance our feeling is that this wording does essentially capture the point the reviewer is making above (which we agree is important.)

1.2

I found this response particularly interesting, and thank the authors for such a detailed reply. Overall, I am happy with the result that this analysis did not change the biological finding in the paper. Personally I found this discussion very useful from a technical standpoint, but do appreciate that using just these p-values as the main results would add confusion. I would be in favour of adding this description and table R1 somewhere to the supplementary materials for interested readers (as such host/pathogen studies are likely to become more common), but leave this decision to the authors.

I also note that there are implementations of linear mixed models which have Bernoulli distributed errors (with logit or probit link) which could be used to further test the theory that differences are down to model misspecification, but I certainly am not asking the authors to do this analysis here, as they have answered my question thoroughly already.

We have included a reformatted version of Table R1 and the associated discussion as a new Supplementary Text section (section 3.3), which we conclude by indicating that caution may be needed when interpreting linear mixed model results for binary phenotypes (or pathogen genotypes treated as an outcome variable) if strong effects are present.

1.4

Thank you for looking into a circo plot / epistasis analysis, and sorry that it was a dead end. Figure 4 is easier to understand now.

John Lees

Referee #2 (Remarks to the Author):

I am completely satisfied with the thorough response of the authors to my comments and those of the other reviewers.

Thank you for your comments.

Referee #4 (Remarks to the Author):

Most of the comments from the initial review have been adequately addressed; however, the authors should consider making it explicit for the reader that figure 3 does not show individual-level comparisons. In saying that the comparison is "ecological", it was meant that the authors should make it clear that although the populations with high HbS frequency also tend to have high frequency of the parasite alleles of interest, this comparison does not indicate that the HbS individuals were more likely to harbor parasites with the alleles of interest.

We have updated the legend to state that this is a population-level comparison, writing:
“Points show the estimated population-level Pfsa+ allele frequency (y axis, as in panel a and b) against HbS allele frequency (x axis) in populations from MalariaGEN Pf6. [...] Pfsa+ allele frequencies were computed from the relevant genotypes, after excluding mixed or missing genotype calls. HbS allele frequencies were computed from frequency estimates previously published by the Malaria Atlas Project¹⁷ within each country, by averaging over the locations of MalariaGEN Pf6 sampling sites weighted by the sample size.” We hope these changes help clarify that the Pfsa and HbS frequencies are computed from separate datasets and are not compared within individual infections in this figure.